# Atmospheric teleconnection processes linking winter air stagnation and haze extremes in China with regional Arctic sea ice decline

Yufei Zou[1], Yuhang Wang[2], Zuowei Xie[3], Hailong Wang[1], Philip J. Rasch[1]

[1]Atmospheric Sciences and Global Change Division, Pacific Northwest National Laboratory, Richland, WA 99354, USA

[2]School of Earth and Atmospheric Sciences, Georgia Institute of Technology, Atlanta, GA 30332, USA

[3]International Center for Climate and Environment Sciences, Institute of Atmospheric Physics, Chinese Academy of Sciences, Beijing, 100029, China

*Correspondence*: Yufei Zou (yufei.zou@pnnl.gov) and Yuhang Wang (yuhang.wang@eas.gatech.edu)

**Abstract.** Recent studies suggested significant impacts of boreal cryosphere changes on wintertime air stagnation and haze pollution extremes in China. However, the underlying mechanisms of such a teleconnection relationship remains unclear. Here we use the Whole Atmosphere Community Climate Model (WACCM) to investigate dynamic processes leading to atmospheric circulation and air stagnation responses to Arctic sea ice changes. We conduct four climate sensitivity experiments by perturbing sea ice concentrations (SIC) and corresponding sea surface temperature (SST) in autumn and early winter over the whole Arctic and three sub-regions in the climate model. The results indicate distinct responses in circulation patterns and regional ventilation to the region-specific Arctic changes, with the largest increase of both the probability (by 132%) and the intensity (by 30%) of monthly air stagnation extremes being found in the experiment driven by SIC and SST changes over the Pacific sector of the Arctic (the East Siberian and Chukchi Seas). The increased air stagnation extremes are mainly driven by an amplified planetary-scale atmospheric teleconnection pattern that resembles the negative phase of the Eurasian (EU) pattern. Dynamical diagnostics suggest that convergence of transient eddy forcing in the vicinity of Scandinavia in winter is largely responsible for the amplification of the teleconnection pattern. Transient eddy vorticity fluxes dominate the transient eddy forcing and produce a barotropic anticyclonic anomaly near Scandinavia and wave-train propagation across Eurasia to the downstream regions in East Asia. The piecewise potential vorticity inversion analysis reveals that this long-range atmospheric teleconnection of Arctic origin takes place primarily via the middle and upper troposphere. The anomalous ridge over East Asia in the middle and upper troposphere worsens regional ventilation conditions by weakening monsoon northwesterlies and enhancing temperature inversions near the surface, leading to more and stronger air stagnation and pollution extremes over eastern China in winter. Ensemble projections based on state-of-the-art climate models in the Coupled Model Intercomparison Project Phase 6 (CMIP6) corroborate this teleconnection relationship between high-latitude environmental changes and middle-latitude weather extremes, though the tendency and magnitude vary considerably among each participating model.

# 1 Introduction

The severe air pollution problem in China has drawn broad attention because of its profound public health (Kan et al., 2012), socioeconomic (Xie et al., 2016), and climatic impacts (Li et al., 2016). In response to an increasing health burden and social costs caused by these environmental stresses, China has prioritized environment protection by implementing unprecedentedly stringent air pollution control policy (the State Council of China, 2013) and achieved great success with gradually decreasing annual mean fine particle (PM$_{2.5}$: particulate matter with aerodynamic diameters of 2.5 micrometers or less) concentrations in recent years (Zhang et al., 2019). However, severe haze pollution associated with PM$_{2.5}$ and PM$_{10}$ (particulate matter with aerodynamic diameters of 10 micrometers or less) in boreal winter still makes clean air a great challenge for China, especially over the East China Plains (ECP) area (Song et al., 2017). Many studies have investigated possible causes of China's haze pollution from various perspectives: massive primary pollution emissions (Liu et al., 2016; Sun et al., 2016), rapid secondary pollution formation (Cheng et al., 2016; Huang et al., 2014; Guo et al., 2014; Wang et al., 2016), unfavorable regional circulation features (Jia et al., 2015; Niu et al., 2010; Yin and Wang, 2017), and positive aerosol-weather feedback effects (Ding et al., 2016; Lou et al., 2019; Zhang et al., 2018; Zhong et al., 2018) have all been identified as contributing factors. An et al. (2019) provides a recent comprehensive review of the severe haze problem in China and emphasizes the synergy among these contributing factors.

It has also been reported that climate change plays an important role in generating conducive meteorological conditions for the favorable formation and unfavorable ventilation of air pollutants in China and many other regions (Cai et al., 2017; Dawson et al., 2014; Hong et al., 2019; Horton et al., 2014; Wang and Chen, 2016). Several possible climate factors have been investigated for their effects on winter haze pollution in China, including changes in: 1) Arctic sea ice (Wang et al., 2015; Zou et al., 2017); 2) Eurasian snow cover (Yin and Wang, 2018; Zou et al., 2017); 3) El Niño–Southern Oscillation (ENSO; Chang et al., 2016; Sun et al., 2018; Zhao et al., 2018; Zhang et al., 2019); 4) Pacific Decadal Oscillation (PDO; Zhao et al., 2016); and 5) northwestern Pacific sea surface temperature (SST; Pei et al., 2018). In those studies, researchers mainly focused on the relationships of various climate factors with pollution-related weather conditions such as the intensity of East Asia winter monsoon (EAWM), planetary boundary layer height, precipitation, and circulation patterns that correlated with winter haze pollution in China. The data analysis results have been further corroborated by modeling studies (Zhao et al., 2016; Zhao et al., 2018; Zhang et al., 2019; Zou et al., 2017). However, a clear understanding of key dynamic processes linking complex meteorological changes to critical climate factors is still missing. This is necessary to establish a robust causal relationship between remote climate drivers and localized atmospheric responses, because a correlation does not necessarily imply causation. Other studies have examined future projections of air stagnation and pollution conditions based on different climate scenarios and ended up with contradictory conclusions over the eastern China region (Cai et al., 2017; Hong et al., 2019; Horton et al., 2014), which further highlights the importance of physical process-based analysis of modeling results.

Given the increasing evidence that climate change—especially that occurring in high-latitude regions—may have an influence on middle-latitude circulation and weather extremes (Cohen et al., 2014; IPCC, 2019), it is imperative to identify the key atmospheric processes driving the circulation

responses and to understand the underlying physical mechanisms for specific extreme weather events.
Several possible dynamic pathways linking Arctic warming to midlatitude weather extremes have been
proposed and investigated in the past few years (Barnes and Screen, 2015; Overland et al., 2016).
However, the observational data and modeling results are sometimes contradictory and are open to
different interpretations (Cohen et al., 2020). Therefore, here we revisit the particular linkage between
Arctic sea ice and wintertime air stagnation in China identified by our previous study (Zou et al., 2017)
and elucidate a teleconnection mechanism based on new climate model sensitivity experiments and
dynamic diagnoses. We describe the analytical methods and datasets in Sect. 2 and analysis of model
results in Sect. 3, which is followed by discussion and conclusions in Sect. 4.
**2 Analysis methods and datasets**
**2.1 Observation and reanalysis data**
Monthly gridded Arctic sea ice concentration (SIC) and sea surface temperature (SST) data for 1950–
2018 were collected from the Met Office Hadley Centre (HadISST; Rayner et al., 2003) for statistical
analysis and comparison with numerical simulation results. We conducted trend analysis for Arctic sea
ice changes and examined the statistical correlation between these changes and key atmospheric
circulation patterns of interest. The National Centers for Environmental Prediction and National Center
of Atmospheric Research (NCEP/NCAR) reanalysis data (Kalnay et al., 1996) was used to calculate
indices of a hemispheric-scale Eurasian (EU) pattern (Wallace and Gutzler, 1981) and a regional
circulation pattern (MCA_Z500) over East Asia in the 500 hPa geopotential height field (Z500) (Fig. S1
in the Supplement). We focused on these two circulation patterns at different spatial scales given their
considerable impacts on winter synoptic weather (Liu et al., 2014; Wang and Zhang, 2015) and regional
haze pollution in China (Li et al., 2019). We followed the definition of the EU index (EUI) in Wallace
and Gutzler (1981) and calculated the EU index in winter (December-January-February; DJF) from
1951 to 2019 (years are aligned with January of the winter season in this work),
$$EUI = -\frac{1}{4}Z^*(55° N, 20° E) + \frac{1}{2}Z^*(55° N, 75° E) - \frac{1}{4}Z^*(40° N, 145° E) \qquad , \qquad (1)$$
where $Z^*$ denotes the normalized monthly mean geopotential height anomalies at 500 hPa using the
1981-2010 average as the climatology. We then regressed the 500 hPa geopotential height anomalies
onto this index to get the EU spatial pattern (Fig. 1c/d), which resembles those reported in Liu et al.
(2014), Wallace and Gutzler (1981), and Wang and Zhang (2015) quite well.
We then used the NCEP/NCAR reanalysis data to calculate gridded pollution potential index (PPI) as
a synthetic meteorological proxy for describing regional air stagnation severity (Zou et al., 2017). The
monthly PPI in winter (DJF) of 1951–2019 was calculated using Eq. (2) as a weighted average of
normalized surface wind speed index (WSI) and near surface air temperature gradient index (ATGI)
based on the reanalysis data. WSI was standardized by subtracting time-averaged climatological mean
of near-surface wind speed over the 1981-2010 period from the monthly values at each grid cell and
then dividing by its standard deviations in the same period. ATGI was the standardized potential
temperature gradient field between 925 and 1000 hPa using the same method. These two indices are
used to reflect horizontal and vertical dispersions of near-surface air pollutants, respectively. We then
estimated grid-scale PPI by weighted averaging WSI and ATGI,
$$PPI = \frac{r_1 \times WSI + r_2 \times ATGI}{|r_1| + |r_2|} \quad , \qquad (2)$$
where $r_1$ and $r_2$ are the Pearson correlation coefficients of WSI ($r_1 = -0.73$) and ATGI ($r_2 = 0.70$) with in
situ $PM_{10}$ observations over the ECP area (Zou et al., 2017). Regional averaged ECP_PPI was estimated
by averaging grid-scale PPI over the ECP area (112° E to 122° E, 30° N to 41° N).
7       Lastly, we applied the Maximum Covariance Analysis (MCA) method (Wilks, 2011) as in our
previous study (Zou et al., 2017) to the Z500 and PPI fields and identified the regional MCA_Z500
pattern that had the largest covariance with PPI changes in ECP. The MCA analysis performs a singular
value decomposition of the covariance matrix of the selected two variables and generates a series of
coupled modes in space and time dimensions for both variables (Wilks, 2011). We chose the first
couple of modes in the Z500 and PPI fields as the MCA_Z500 and MCA_PPI patterns that show the
largest covariance with each other ($r$=0.65; Fig. S1b in the Supplement). The MCA_Z500 pattern
resembles a regional manifestation of the planetary-scale EU pattern (in negative phase) with a good
correlation between these two indices ($r$=-0.67; Fig. S1b in the Supplement). However, it's worth noting
that this regional MCA_Z500 pattern can also be excited by other large-scale teleconnection processes
such as the East Atlantic pattern or the East Atlantic/Western Russia pattern associated with both
natural variability and perturbed Rossby wave activity (Lim, 2015; Simmons et al., 1983). These
variables are assessed as metrics of circulation and ventilation responses to climate forcing in the
following sections.
**2.2 Climate models and numerical sensitivity experiments**
This study uses the high-top Whole Atmosphere Community Climate Model (WACCM) version 5
(Marsh et al., 2013) within the common numerical framework of the NCAR Community Earth System
Model (CESM) used for climate sensitivity experiments. WACCM is a comprehensive atmospheric
model with a well-resolved stratosphere of 70 vertical layers spanning the surface to the thermosphere
(~0.001 hPa) at a horizontal resolution of 1.9° (latitude) ×2.5° (longitude). We conducted 30-year
simulations (with an additional unanalyzed 1-year period for the control run as spin-up) as the control
(CTRL) run with annually repeating prescribed climatological (1981-2010 average) Arctic SIC and SST
from the Met Office Hadley Centre (Rayner et al., 2003) (Table 1). We then performed four climate
sensitivity experiments by perturbing SIC and SST in different Arctic regions to investigate the climate
sensitivity to regional Arctic sea ice changes and associated local ocean warming (Screen et al., 2013).
The spatial distribution of correlation coefficients between SIC and EU indices (Fig. 1b) reveals
varying climate sensitivity relationships between regional sea ice changes and circulation responses as
suggested by previous studies (Screen, 2017; Sun et al., 2015; McKenna et al., 2018). To test this
region-specific climate sensitivity, we first perturbed SIC and SST in the whole Arctic region to
evaluate their comprehensive climate effects, and then divided the whole Arctic region into three sub-
regions (R1-R3; Fig. 1b) and perturbed regional SIC and SST in three region-specific numerical
experiments (Table 1). Specifically, we branched an 8-month simulation from each July of the CTRL
run with observed SIC/SST data in autumn and early winter (August-November) of 2012 over the
whole Arctic in the first sensitivity experiment (SENSall). We chose 2012 because it had the lowest
level of Arctic sea ice concentrations throughout the satellite era of the last four decades
(NSIDC/NASA, 2019) and thus provides the strongest sea-ice forcing to the climate system. We only
changed the surface boundary conditions (SIC/SST) at modeling grid cells with SIC anomalies larger
than 10% to focus on the Arctic regions with the most significant changes. We then added three region-
specific sensitivity experiments (SENSr1/r2/r3) by perturbing regional SIC and SST in R1-R3 regions
(R1: 30° E to 150° E, 70° N to 85° N; R2: 150° E to 145° W, 60° N to 85° N; R3: 145° W to 30° W,
50° N to 85° N; Fig. 1b), respectively, following the same perturbation method in SENSall.
We analyzed the consecutive December-January-February monthly data at the end of each sensitivity
simulation to examine the seasonal impact of Arctic sea ice changes in comparison with observation and
reanalysis data. The simulated EU/MCA_Z500 circulation indices were estimated by projecting
modeling differences (SENSx-CTRL, x=all/r1/r2/r3) onto the reanalysis-based EU/MCA_Z500
patterns, and the ECP_PPI indices in the model were calculated following the same method of the
reanalysis one by using the CTRL ensemble mean as the climatology.
Since the default 2000-based emission inventory (Lamarque et al., 2010) in WACCM was prepared
for the Fifth Assessment Report of the Intergovernmental Panel on Climate Change (IPCC AR5) and is
low biased over China, we updated the anthropogenic emission inventory for China by replacing the
default one with the 2010-based multi-resolution emission inventory for China (MEIC; Li et al., 2017).
The MEIC-MIX inventory was developed for the years 2008 and 2010 and has been widely used for air
pollution simulation and health impact assessment studies in China (Geng et al., 2017; Zhang et al.,
2017). It is worth noting that the main objective of this study is not to reproduce severe haze pollution
extremes in China, but to understand how regional atmosphere and pollution conditions respond to the
key climate drivers in the high latitudes. Therefore, we only focused on the relative changes of PPI and
surface $PM_{2.5}$ concentrations between SENS and CTRL experiments and investigated dynamic
processes associated with these changes in our following analysis.
Besides the CESM-WACCM model used in the sensitivity experiments, we also analyzed results
from other state-of-the-art climate models in the latest CMIP6 project to examine the teleconnection
relationship between Arctic sea ice and regional air stagnation in China. Table S1 in the Supplement
lists the 8 CMIP6 models with the same experiment and variant ID (r1i1p1f1) used for historical
simulations and future projections of the Arctic sea ice extent (SIE) and ECP_PPI time series. SIE is a
measurement of the ocean area where sea ice concentrations exceed 15% (NSIDC, 2019). We analyzed
historical simulations (1950-2014; Eying et al., 2016) and future projections (2015-2100) of the Shared
Socioeconomic Pathway under a high greenhouse gas emission scenario (SSP5-8.5; O'Neill et al.,
2016) by each model to maintain consistency with previous studies (Cai et al., 2017; Horton et al.,
2014). We then calculated time series of regional averaged Arctic SIE and ECP_PPI in each CMIP6
model to estimate ensemble means and standard deviations of these variables. The estimation of the SIE
relative changes and ECP_PPI indices in CMIP6 followed the same method of the observation- and
reanalysis-based ones by using 1981-2010 historical runs as the climatology. The whole 150-year
CMIP6 time series was equally divided into three time periods (P1-P3) to evaluate regional air
stagnation conditions under different Arctic sea ice forcing.

## 2.3 Statistical analysis methods

We examined long-term linear trends of observed SIC in each grid cell in Fig. 1a using the Mann-Kendall test, a non-parametric (i.e., distribution free) method that is based on the relative ranking of data values. After trend detection, we estimated the Pearson correlation between the gridded sea ice variations and the EU index. To evaluate the circulation impact on regional ventilation, we conducted composite analysis of gridded PPIs over the middle latitude regions (Fig. 1b) and examined their statistical significance using the two-sided Student's t-test. The t-test was also used to evaluate statistical significance of surface heat flux changes and atmospheric responses in the modeling results, such as the ensemble mean differences of atmospheric variables between the WACCM SENS and CTRL experiments.

To further evaluate the modeling sensitivity results, we used statistical functions in the Python SciPy v1.4.1 module (https://docs.scipy.org/doc/scipy/reference/stats.html; last access: 24 October, 2019) to evaluate statistical properties of the modeling samples and estimate their cumulative distribution functions (CDFs) and probability density functions (PDFs) following proper distributions. We first examined the statistics of the MCA_Z500 and ECP_PPI indices in each experiment in terms of their location, scale, and shape (Table S2), and then conducted the Shapiro-Wilk normality test (Wilks, 2011; the "shapiro" function in SciPy v1.4.1) to examine whether the data conform to normal distributions (Fig. S2 and Fig. S3 in the Supplement). The null hypothesis of the normality test is that the sampling data from each experiment are drawn from a normal distribution. If the p-value is larger than 0.05, we failed to reject the null hypothesis and fitted normal distribution (the "norm.fit" function in SciPy v1.4.1) CDF/PDF curves to the modeling data (30 years × 3 winter months = 90 samples). Otherwise, we rejected the null hypothesis and chose a proper non-Gaussian distribution (e.g., the "gumble_r.fit"/"gumble_l.fit" functions in SciPy v1.4.1 for right-skewed/left-skewed distributions) to fit CDF/PDF curves to the data. The autocorrelation between consecutive months in each experiment is minimal, suggesting mutually independent sampling variables for statistical analysis. Table S2 shows statistical properties and test results of each experiment that indicate the data in most experiments conform to normal distributions except MCA_Z500 in SENSall and MCA_Z500/ECP_PPI in SENSr2. The statistics and histograms of the SENSall MCA_Z500 indices suggest a skew distribution to the left, while those of the SENSr2 MCA_Z500/ECP_PPI indices suggest a skew distribution to the right. Therefore, we fitted a left-skewed Gumbel distribution to the SENSall MCA_Z500 data and a right-skewed Gumbel distribution to the SENSr2 data, respectively. The goodness-of-fit results are shown in the Q-Q plots of Fig. S2 and Fig. S3 in the Supplement. After distribution fitting, we chose the 95[th] percentiles of the MCA_Z500 and ECP_PPI indices in CTRL as the thresholds of positive extremes and estimated the probability of positive extremes in the four SENS experiments based on their fitted CDF curves (i.e., $P_{PPI_{SENS} \geq PPI_{CTRL}^{95th}}$). The average intensity of positive extreme values in each experiment was estimated by weighted averaging values with their probabilities as weights. The fitted CDFs for all the WACCM experiments are shown in Fig. 3 and discussed below. For CMIP6 data, we used the same approach to fit CDF curves for each modeling and the reanalysis data in different time periods. The CDFs for three time periods over 1950-2000 (P1), 2001-2050 (P2), and 2051-2100 (P3) are shown in Fig. 7 and discussed later near that figure. The P1 time period over 1951-2000 was chosen as the reference period for the NCEP reanalysis and CMIP6 modeling data. The thresholds of positive

extremes in the reanalysis and CMIP6 models were defined as the 95[th] percentiles of ECP_PPI values in
this reference time period, which were then used to evaluate probability changes of positive extremes in
the other two periods (i.e., $P_{PPI_{P2/3} \geq PPI_{P1}^{95th}}$).
Such extreme value analyses provide an alternative perspective to the traditional ensemble mean
statistical analysis, lending a more comprehensive understanding of atmospheric responses to climate
forcing based on full distribution curves. A special report of the Intergovernmental Panel on Climate
Change (IPCC, 2012) focusing on the risks of climate extreme events discussed three kinds of responses
including "shifted mean", "increased variability", and "changed symmetry" in climate variable
distributions to climate change. These distinct responses demonstrate that changes in extremes can be
linked to changes in the mean, variance, and shape of probability distributions (IPCC, 2012). We
followed this analysis framework to examine statistical distribution changes in regional circulation
(MCA_Z500) and ventilation (ECP_PPI) with consideration of both natural variability and perturbation-
induced responses in our climate sensitivity experiments. The uncertainty of the extreme probabilities
and intensities in each experiment was estimated using 95% percentile ranges (i.e., values between the
2.5[th] percentile to 97.5[th] percentile) via a bootstrap method by resampling the model simulated samples
with replacement (10000 times) and re-estimating those statistics based on the repeatedly fitted CDFs
(Tables S3 and S4 in the Supplement).
**2.4 Diagnostics of atmospheric dynamics**
To understand the atmospheric pathways from the Arctic sea ice forcing to regional circulation
responses, we employed multiple dynamic diagnostic tools to investigate storm-track characteristics and
local interactions between transient eddy forcing and the time-mean flow. The properties of transient
eddies were depicted by eddy kinetic energy (EKE) in Eq. (3) and the horizontal components of
extended Eliassen-Palm vectors (**E** vectors) in Eq. (4) given by Trenberth (1986),
$EKE = \frac{1}{2}\left(\overline{u'^2} + \overline{v'^2}\right)$           ,                                            (3)
$\boldsymbol{E}\ vector = \frac{1}{2}\left(\overline{v'^2} - \overline{u'^2}\right)\mathbf{i} - \overline{u'v'}\mathbf{j}$           ,                                            (4)
where $u$ and $v$ are the daily zonal and meridional wind components, respectively. The prime denotes the
2–8-day band-pass-filtered quantities and the overbar denotes temporal averaging over a month.
The direction of **E** vectors approximately points to the wave energy propagation relative to the local
time-mean flow, while the divergence and curl of **E** vectors indicate eddy-induced acceleration of local
mean zonal and meridional winds (Trenberth, 1986).
We then illustrated transient eddy feedback to the quasi-stationary flow by eddy-induced
geopotential height tendencies due to the convergence and divergence of transient eddy vorticity and
heat fluxes (Lau and Holopainen, 1984; Lau and Nath, 1991),
$\left\{\frac{1}{f}\nabla^2 + f\frac{\partial}{\partial p}\left(\frac{1}{\sigma}\frac{\partial}{\partial p}\right)\right\}\left(\frac{\partial \phi}{\partial t}\right) = D^V + D^H,\ where\ D^V = -\nabla\cdot(\overline{V'\zeta'})\ and\ D^H = f\frac{\partial}{\partial p}\left(\frac{\nabla\cdot(\overline{V'\theta'})}{S}\right).$     (5)
In Eq. (5), $D^V$ and $D^H$ are the eddy forcing due to heat and vorticity fluxes, respectively. $f$ is the
Coriolis parameter, $\phi = gz$ is geopotential, $\sigma = -\left(\frac{\alpha}{\theta}\right)\left(\frac{\partial \theta}{\partial p}\right)$ is static stability, $\alpha$ is specific volume, $\theta$ is
potential temperature with $S = -\frac{\partial \theta}{\partial p}$, $V$ is horizontal wind, and $\zeta$ is relative vorticity. Here the prime

and overbar are the same with those in Eqs. (3) and (4). By inverting the eddy forcing terms $D^V$ and $D^H$ on its right-hand side separately and solving the equation, we could distinguish independent effects of vorticity and heat fluxes induced by transient eddies on the corresponding height tendencies $Z_t^V$ and $Z_t^H$. The net tendency associated with the combination of $D^V$ and $D^H$ is denoted as $Z_t^{V+H}$.

Moreover, we used the phase-independent 3-dimensional wave activity flux (WAF; Takaya and Nakamura, 2001) based on the monthly averaged reanalysis and modeling data to diagnose zonal and vertical propagation of locally forced wave packet induced by quasi-geostrophic (QG) eddy disturbances embedded in a zonally varying basic flow,

$$\mathbf{W} = \frac{p\,cos\emptyset}{2|\mathbf{U}|} \begin{bmatrix} \frac{u}{a^2 cos^2\emptyset}\left[\left(\frac{\partial\psi'}{\partial\lambda}\right)^2 - \psi'\frac{\partial^2\psi'}{\partial\lambda^2}\right] + \frac{v}{a^2 cos\emptyset}\left[\frac{\partial\psi'}{\partial\lambda}\frac{\partial\psi'}{\partial\emptyset} - \psi'\frac{\partial^2\psi'}{\partial\lambda\partial\emptyset}\right] \\ \frac{u}{a^2 cos^2\emptyset}\left[\frac{\partial\psi'}{\partial\lambda}\frac{\partial\psi'}{\partial\emptyset} - \psi'\frac{\partial^2\psi'}{\partial\lambda\partial\emptyset}\right] + \frac{v}{a^2}\left[\left(\frac{\partial\psi'}{\partial\emptyset}\right)^2 - \psi'\frac{\partial^2\psi'}{\partial\emptyset^2}\right] \\ \frac{f_0^2}{N^2}\left\{\frac{u}{a\,cos\emptyset}\left[\frac{\partial\psi'}{\partial\lambda}\frac{\partial\psi'}{\partial z} - \psi'\frac{\partial^2\psi'}{\partial\lambda\partial z}\right] + \frac{v}{a}\left[\frac{\partial\psi'}{\partial\lambda}\frac{\partial\psi'}{\partial z} - \psi'\frac{\partial^2\psi'}{\partial\lambda\partial z}\right]\right\} \end{bmatrix} + \mathbf{C}_U M \qquad . \qquad (6)$$

Here $u$ and $v$ are the zonal and meridional wind components, respectively. $\mathbf{U} = (u, v, 0)^T$ is a steady zonally inhomogeneous basic flow. $p = (\text{pressure}/1000 \text{ hPa})$ is normalized pressure, $\psi'$ is a streamfunction perturbation relative to the climatological mean, $(\emptyset, \lambda)$ are latitude and longitude, $a$ is the earth's radius, $N^2 = (R_a p^\kappa / H)(\partial\theta/\partial z)$ is the squared buoyancy frequency, $\mathbf{C}_U$ represents the phase propagation in the direction of $\mathbf{U}$, and M can be interpreted as a generalization of small-amplitude pseudo-momentum for QG eddies onto a zonally varying basic flow.

Lastly, we quantified the influence of circulation anomalies at different vertical levels using a piecewise potential vorticity (PV) inversion method (Black and McDaniel, 2004; Xie et al., 2019). The PV anomalies were calculated with reanalysis and simulation data for all troposphere pressure levels from 1000 hPa to 100 hPa in Eq. (7),

$$q' = \frac{1}{f}\left[\frac{1}{(a\,cos\phi)^2}\frac{\partial^2}{\partial\lambda^2} + \frac{f}{a^2 cos\phi}\frac{\partial}{\partial\phi}\left(\frac{cos\phi}{f}\frac{\partial}{\partial x}\right) + f^2\frac{\partial}{\partial p}\left(\frac{\partial}{\partial p}\right)\right]\Phi' \qquad , \qquad (7)$$

where $q$ is the PV, $\phi$ is the geopotential, $f$ is the Coriolis parameter, and a prime represents the deviation from the smoothed climatological annual cycle. We then inverted individual PV "pieces" at different levels to evaluate low-level (850 hPa) horizontal wind anomalies related to these PV anomalies. The horizontal anomalous wind field that will be presented in Fig. 6 was derived from the geopotential height field based on geostrophic balance. We partitioned the 1000-100 hPa PV anomalies into the lower (1000-850 hPa) and the middle to upper troposphere (700-100 hPa) PV anomalies and compared their impacts on the low-level wind field in Sect. 3.

## 3 Results

### 3.1 Observation- and reanalysis-data based relationships among Arctic sea ice, atmospheric circulation, and boundary-layer ventilation

We first examine the long-term variations of Arctic sea ice in autumn and early winter (August to November, ASON) of the past four decades during the satellite era. Figure 1a shows strong decreasing trends in Arctic SIC, especially in the Eurasian and Pacific sectors such as the northern Barents Sea,

Kara Sea, East Siberian Sea, and Chukchi Sea. The winter EUI shows positive correlations with regional Arctic sea ice concentrations with the strongest correlation over the East Siberian Sea and Chukchi Sea (Fig. 1b), suggesting a decrease of EUI in winter following the sea ice decline over these regions in preceding months. The correlation coefficient between EU and regional averaged SIC over these positively correlated R2 grids is 0.38 (p=0.02). Positive correlations are also present when the long-term trend in regional sea ice changes is removed, suggesting a consistent relationship between EU variations and sea ice changes in these Arctic regions on both long-term (interdecadal) and short-term (interannual) time scales.

To evaluate the impact of EU phases on regional ventilation, we conducted composite analysis and compared wintertime boundary-layer PPI differences over the Northern Hemisphere corresponding to different EU phases. In general, PPI and EU show an in-phase relation with high (low) PPI anomalies corresponding to positive (negative) height anomalies in the EU pattern (Fig. 1c, d). Europe and East Asia become two hot-spot regions in the negative EU phase (Fig. 1d), implying significant sensitivity (i.e., lower ventilation capability and higher air pollution potential) in these regions. Since the EUI shows a positive correlation with declining sea ice in the Pacific sector of the Arctic, we would expect more severe air stagnation over East Asia coinciding with the decrease of EUI and regional Arctic sea ice.

## 3.2 WACCM sensitivity simulations

The statistical analysis suggests a potential linkage between the Arctic sea ice decline and the regional ventilation deterioration through a circulation change in the negative EU phase. We evaluate this teleconnection relationship using ensemble WACCM sensitivity experiments (Table 1). Figure 2 shows the mean surface sensible plus latent heat flux changes between the SENSall and CTRL experiments during autumn and winter. Since we perturbed the Arctic SIC and SST in the model surface boundary conditions from August to November, most Arctic regions show significantly increased heat fluxes in autumn and early winter, especially over the Kara Sea, Laptev Sea, East Siberian Sea, and Beaufort Sea (Fig. 2a). The heat flux changes are much weaker in winter with some remnant influence over the Kara Sea (Fig. 2b) due to the strong perturbation in this region. The comparison of monthly variations in regional averaged heat fluxes over the Arctic confirms the much stronger forcing in ASON (seasonal mean heat fluxes increased by 3.3 W/m$^2$ or +16% over the Arctic in SENSall) than in DJF (seasonal mean heat fluxes decreased by 0.17 W/m$^2$ or -1% over the Arctic in SENSall) (Fig. 2c).

To examine the regional circulation and ventilation responses to these changes in the high latitudes, we fit the CDF and PDF curves of MCA_Z500 and ECP_PPI based on CTRL and SENS monthly results in winter. Figure 3 shows the CDF changes of simulated MCA_Z500 (Fig. 3a) and ECP_PPI indices (Fig. 3b) between sensitivity and CTRL experiments. It is clear that both indices show more significant changes in their extreme members than in medians or ensemble means, especially in SENSr2 driven by SIC and SST changes in the Pacific sector of the Arctic (R2 in Fig. 1b). In SENSr2, the occurrence probability of MCA_Z500 positive extremes increases by 50% from 5.0 to 7.5% (95[th] percentile range: 0.8-16.4%) (Fig. 3a; Table S3 in the Supplement), while the ECP_PPI positive extremes increases by 132% to 11.6% (95% percentile range: 5.2-18.4%) (Fig. 3b; Table S3 in the Supplement). Meanwhile, the intensity of positive extreme values of the two indices also increases by

11% and 30%, respectively (Table S4 in the Supplement). Only SENSr2 shows statistically significant increases of ECP_PPI in terms of positive extreme probability and intensity, and the significance of such increases is independent of the fitting method being used (i.e., still valid with nonparametric curve fitting). In contrast, the changes of MCA_Z500 in all experiments are not statistically significant, which might be attributable to the need for larger ensemble sizes to detect dynamically-modulated responses (e.g., SLP and geopotential height anomalies) as compared to thermally-directed responses (e.g., vertical temperature gradient anomalies and their effects on PPI) (Screen et al., 2014). The substantially increased ECP_PPI positive extremes in SENSr2 contribute to the positive responses in its ensemble mean, making SENSr2 the only sensitivity experiment with positive ensemble mean ECP_PPI (0.03, not statistically significant). In comparison, other SENS experiments generally show negative ensemble mean ECP_PPI values due to left-shifted CDF curves at most percentiles. For instance, SENSr1 is the only experiment showing robustly decreased ECP_PPI at all percentiles in its CDF curve (Fig. 3b), contributing to its negative ensemble mean of ECP_PPI (-0.13) that is statistically significant at the 0.05 significance level (Table S2 in the supplement). This result implies an overall improvement of the ECP regional ventilation driven by the SIC and SST changes in the Barents-Kara Seas (R1 in Fig. 1b), while the ventilation responses are more erratic driven by sea ice loss in other Arctic regions.

The disparate results among the sensitivity experiments highlight distinct climate effects of regional SIC/SST changes as suggested by both statistical analysis in the last section and previous climate modeling studies. Screen (2017) and McKenna et al. (2018) investigated atmospheric responses to regional sea ice loss by perturbing regional SIC and SST or surface temperature. McKenna et al. (2018) focused on the climate impacts of sea ice loss in the Atlantic (the Barents-Kara Seas) and the Pacific sectors (the Chukchi-Bering Seas) of the Arctic, while Screen (2017) conducted more comprehensive investigation by dividing the whole Arctic region into nine sub-regions. These region-specific modeling studies suggested quite different or even opposite effects of regional sea ice forcing on general circulation in the stratosphere and troposphere. However, it is worth noting that they mainly focused on the responses in the stratospheric polar vortex and in the tropospheric Arctic Oscillation (AO) and North Atlantic Oscillation (NAO), which are different from the EU and MCA_Z500 patterns of interest in this work.

The differences in the MCA_Z500 and ECP_PPI responses among the four sensitivity experiments in extreme members and ensemble means also suggest complex relationships between Arctic sea ice loss and mid-latitude weather changes. Two distinct patterns of Asian winter climate responses to Arctic sea ice loss were identified in a previous study (Wu et al., 2015), one (the "Siberian High" pattern) in positive phase associated with the strengthened Siberia High and EAWM systems while the other (the "Asia-Arctic" pattern) in negative phase associated with weakened EAWM and enhanced precipitation in East Asia. Such opposite responses in regional climate and weather systems partly explain the concurrent changes in the two tails of distribution curves in our sensitivity experiments. An IPCC report (IPCC, 2012) demonstrated three kinds of responses in variable probability distributions to climate change, and our ECP_PPI results in the four SENS experiments agree with the proposed "increased variability", "shifted mean", "changed symmetry", and "increased variability" responses, respectively (see section 2.3 and Table S2 and Fig. S3 in the Supplement for explanations). These distinct responses reflect complex interactions between atmospheric anomalies driven by climate forcing and atmospheric circulation associated with the natural variability.

Coupling processes among different components of the climate system can compound such complexity by amplifying or dampening signal-to-noise ratios and expanding responsive regions (Deser et al., 2015; Deser et al., 2016). Smith et al. (2017) pointed out the importance of ocean-atmosphere coupling and the background state in modulating atmosphere responses to Arctic sea ice changes. They found that the background state plays a key role in determining the sign of the NAO responses to Arctic sea ice loss via the refraction of planetary waves by the climatological flow (Smith et al., 2017). These findings shed light on the diverse responses in our simulated distributions of MCA_Z500 and ECP_PPI because of the varying background flow in each modeling year. It is worth noting that we only changed the surface boundary conditions (SIC/SST) at these model grid cells with sea ice changes larger than 10% and kept other grid cells unperturbed. Because our modeling study used prescribed ocean data, the ice-ocean-atmosphere coupling is constrained and it might attenuate other atmospheric responses to sea ice changes as discussed in previous studies (Deser et al., 2015; Deser et al., 2016; Smith et al., 2017). Exacerbated haze pollution represented by positive anomalous $PM_{2.5}$ surface concentrations in eastern China regions appears concurrently with increased ECP_PPI extreme values in all SENS experiments. The changes in surface $PM_{2.5}$ concentration fields correspond well with the PPI changes with the most significant increases in SENSr2 (Fig. S4c in the Supplement) due to the largest number of positive extreme members in this case (Fig. 3b). These positive PPI extremes in SENSr2 are attributable to both reduced surface wind speed and enhanced near surface temperature inversion (Fig. S5 in the Supplement). The agreement between PPI and surface $PM_{2.5}$ concentration changes demonstrates the promising capability of PPI for describing regional air stagnation and pollution potentials. Since the cause of increasing air stagnation extremes is the major concern of this work, we conduct more dynamic diagnosis in the next section to understand the following two questions: (1) how does severe air stagnation occur in these SENSr2 extreme members? (2) why are there more and intensified air stagnation extremes in SENSr2?

**3.3 Diagnosis of dynamic processes**

To answer the first question raised in the previous section, we examine atmospheric anomalies induced by sea ice perturbations in the SENSr2 extreme members from the perspective of wave activity fluxes and transient eddy feedback forcing. We also evaluate climate impacts of the anomalous circulations and teleconnection patterns in these extremes on regional ventilation using the piecewise PV inversion method. We first compare the anomalous geopotential height field in the upper troposphere (250 hPa, Fig. 4a) and sea level pressure anomalies (Fig. 4b), both of which share similar features with strong positive anomalies over the North Pacific and Northern Europe and negative ones over central Siberia. This quasi-barotropic structure over most regions of the Northern Hemisphere agrees with prior findings regarding Arctic sea ice induced atmospheric responses at interseasonal scales (Deser et al., 2010). The geopotential height anomalies in the upper troposphere manifest wave-train patterns with enhanced Rossby wave propagation from the North Pacific to North America and over the Eurasian continent (Fig. 4a). The sea level pressure anomalies exhibit eastward displacements with respect to the upper-troposphere anomalies over Eurasia (Fig. 4b). In particular, the negative sea level pressure anomaly over Siberia extends southeastward to southeastern China, suggesting a weakened Siberia high. In

response, anomalous surface southerlies are seen along the coastal region of eastern China that offset
the prevalent winter monsoon and thus increase the air stagnation in winter over eastern China.
We then compare the difference of EKE and $\mathbf{E}$ vectors (section 2.4) between SENSr2 extreme
members and CTRL ensemble mean and calculated anomalous 250 hPa geopotential height tendencies
driven by transient eddies using Eq. (5). The most prominent features in the anomalous zonal wind and
transient eddy fields are zonal positive anomalies over middle latitudes from the northeastern Pacific to
North Africa that are to the south of the upper-troposphere negative height anomalies (Fig. 4c).
Meanwhile, both zonal wind and EKE fields feature a moderate dipole around the positive height
anomaly. The divergence of $\mathbf{E}$ vectors is seen from the northeastern Pacific to the North Atlantic, which
results in amplifications of zonal winds and a climatology-like pattern of transient eddy feedback
forcing with zonally elongated positive height tendencies to the south of negative height tendencies
(Fig. 4d). In contrast, the $\mathbf{E}$ vectors converge over the vicinity of the Scandinavian region (Fig. 4d),
suggesting a weakened zonal wind and thereby depress transient eddy activity. Accordingly, significant
geopotential height tendencies driven by transient eddy forcing emerge in the upper troposphere,
showing pronounced positive anomalies near the Scandinavian region. These tendencies are dominated
by transient eddy vorticity forcing rather than transient eddy heat forcing, the latter of which shows
opposite but much weaker effects on the upper level geopotential height field (Fig. S6a, b in the
Supplement). Both transient eddy vorticity and heat flux forcing contribute constructively in the lower
troposphere (Fig. S6c, d in the Supplement).
We further compare the simulated atmospheric anomalies relative to the ensemble average of the 30
strongest negative EU years (10 minimums for each Dec/Jan/Feb month) in winter since 1950 in the
reanalysis data. Figure 5 shows the horizontal (250 hPa) and vertical structures of wave propagation in
reanalysis-based negative EU extremes (Fig. 5a, d) and model-based SENSr2 extreme members (Fig.
5b, e; hereafter SENSr2$_{extreme}$) and their CTRL counterparts (Fig. 5c, f; hereafter CTRL$_{counterpart}$). The
comparison between the last two modeling results shed light on the question of why more extremes
occur in SENSr2 than in CTRL. It's apparent that SENSr2$_{extreme}$ shares more similar features of wave
train propagation with the reanalysis data than the CTRL$_{counterpart}$ does, characterized by two anomalous
troughs over the North Atlantic (region A in Fig. 5a, b) and the Siberian (region C in Fig. 5a, b) areas
and two anomalous ridges over the Scandinavian Peninsula (region B in Fig. 5a, b) and East Asia
(region D in Fig. 5a, b) areas. These wave train patterns exhibit barotropic vertical structures in the
troposphere in both cases (Fig. 5d, e). Unlike the reanalysis data, the wave train pattern of SENSr2$_{extreme}$
shows a westward tilt in the lower troposphere with evident downward energy propagation. In contrast,
such prominent configurations fade in CTRL$_{counterpart}$ with much weaker wave activity, resulting in
disappeared key features along the propagation pathway (Fig. 5c, f). Though these CTRL$_{counterpart}$
members share the same initial condition with SENSr2$_{extreme}$, they show different vertical structures of
wave train patterns in both upstream regions (e.g., the lower troposphere over region B) and the
downstream region of East Asia (region D). Meanwhile, the anomalous centers of CTRL$_{counterpart}$
members are higher than SENSr2$_{extreme}$ and the reanalysis data, which is unfavorable for the lateral
Rossby wave propagation to help the formation of the positive height anomalies over East Asia. In
contrast to negative to neutral height anomalies in CTRL$_{counterpart}$, SENSr2$_{extreme}$ manifests positive
anomalies in the middle to upper troposphere over this region (region D in Fig. 5e). To highlight such
difference between SENSr2$_{extreme}$ and CTRL$_{counterpart}$, we isolate the sea ice perturbation-induced
anomalies in SENSr2$_{extreme}$ (shading in Fig. S7c, d in the Supplement) by subtracting CTRL$_{counterpart}$
from SENSr2$_{extreme}$ and overlay them with the internal variability-induced anomalies in CTRL$_{counterpart}$
(contours in Fig. S7d in the Supplement). The sea ice-induced anomalous Rossby wave constructively
interferes with the internal variability-induced one over the upstream regions including Northern Europe
and Central Siberia, contributing to the enhanced wave propagation to downstream regions with
emerging high-pressure anomalies over East Asia in SENSr2$_{extreme}$ (Fig. S7d in the Supplement). This
critical difference appears to be the key to more frequent ECP_PPI extremes in the SENSr2 experiment.
To illustrate this point, we use the piecewise PV inversion method to examine the impact of the
circulation anomalies in region D (the red box in Fig. 5a) on regional ventilation over eastern China in
both the negative EU reanalysis data and modeling results from SENSr2$_{extreme}$ and CTRL$_{counterpart}$. We
first partition tropospheric PV anomalies into two parts: the lower troposphere (1000-850 hPa) and the
middle to upper troposphere (700-100 hPa) and then invert each PV piece at two levels to estimate the
low-level (850 hPa) horizontal anomalous winds associated with these PV anomalies. We find
significantly weakened wind fields in eastern China in the reanalysis data and SENSr2$_{extreme}$, but not in
CTRL$_{counterpart}$. In contrast to strong climatological northwesterly winds over northeastern Asia (Fig. 6a,
b, c), PV anomalies in the middle to upper troposphere in both reanalysis-based negative EU and
model-based SENSr2$_{extreme}$ induce anomalous southeasterly winds in the lower troposphere over the
ECP region (Fig. 6d, e), which is not the case in CTRL$_{counterpart}$ (Fig. 6f). These anomalous
southeasterlies weaken the monsoon northwesterlies and strengthen air stagnation in this region in the
first two cases. We also compare the contribution of PV anomalies at different levels and find that the
ventilation suppression effect is dominated by anomalous PV in the middle to upper troposphere (700
hPa and above) rather than that in the lower troposphere (below 700 hPa) in both reanalysis and
SENSr2$_{extreme}$ data. Comparing to anomalous southerly winds induced by PV anomalies at middle to
upper levels (Fig. 6d, e), those PV anomalies in the lower troposphere mainly tend to strengthen
northerly climatological winds over the ECP region, though the circulation patterns in the reanalysis and
SENSr2$_{extreme}$ results (Fig. 6g, h) are quite different from CTRL$_{counterpart}$ (Fig. 6i). In general, the
ventilation suppression effect associated with middle- and upper-level PV anomalies overwhelms the
enhancement effect associated with lower-level PV anomalies and finally suppresses monsoon winds as
a net effect in the reanalysis and SENSr2$_{extreme}$ results (Fig. 6j, k), while CTRL$_{counterpart}$ manifests an
opposite net effect with the dominant role of lower-level PV anomalies (Fig. 6l).

## 31 3.4 Historical simulations and future projections in CMIP6

Lastly, we examine the historical simulations and future projections of Arctic SIE and ECP_PPI under
the SSP5-8.5 scenario based on 8 currently available CMIP6 climate models (Eying et al., 2016; see
Table S1 in the Supplement for model details) to understand how this teleconnection relationship might
change in the future. Figure 7 shows the time series of the Arctic sea ice and ECP_PPI and the statistical
distribution changes of ECP_PPI among three time periods: P1 (1951-2000) defined as the reference
period with slowly declining Arctic SIE, P2 (2001-2050) as the near-term projection with rapidly
decreasing Arctic SIE, and P3 (2051-2100) as the long-term projection with an almost ice-free Arctic in
boreal Autumn. A similar figure is available in the Supplement to show the relationship between Arctic
SIE change and MCA_Z500 time series (Fig. S8). Although the CMIP6 model ensemble captures the
observed decreasing trend in Arctic SIE, it generally shows less interannual and interdecadal variability
of Arctic SIE and ECP_PPI than the reanalysis data. Low decadal variations in the CMIP6 models are
also evident in the CDF distributions of simulated ECP_PPI. For instance, the simulated ECP_PPI CDF
curve in P1 is positively shifted over the whole distribution range in comparison with the reanalysis-
based one (Fig. 7b), while the simulated CDF curve in P2 is negatively shifted relative to the reanalysis
data (Fig. 7c) especially over the lower ends of the distribution. Therefore, the shift to positive PPI
distributions from P1 to P2 in the NCEP reanalysis data is much more significant than the CMIP6
ensemble, which is understandable since the reanalysis data is just a single realization and the CMIP6
modeling result is ensemble average. This shrunken interdecadal change in the CMIP6 model ensemble
is also found in MCA_Z500 but to a less prominent extent (Fig. S8b-d in the Supplement).
Consequently, the ensemble mean values and averaged probability of simulated ECP_PPI positive
extremes increase from -0.05/5% in P1 to 0.07/7% in P2 (Fig. 7c), which is smaller than those presented
in the reanalysis data (-0.38/5% in P1 to 0.30/19% in P2). With the greatest change of Arctic SIE in P3,
the ensemble averaged probability of ECP_PPI positive extremes nearly doubles and increases to 9%
with the ECP_PPI mean value of 0.09 (Fig. 7d), in concurrence with substantially increased positive
extremes of MCA_Z500 in the same time period (Fig. S8d in the Supplement). The model-specific
projections of ECP_PPI positive extreme probabilities range between 2%-11% in P2, and between 2-
13% in P3 (Table S5 and Fig. S9 in the Supplement).
For more direct comparison with the CESM-WACCM sensitivity results in previous sections, we
specifically looked into the two newer versions of CESM (CESM2 and CESM2-WACCM; Table S1 in
the Supplement) that were developed in the same CESM project as the CESM-WACCM model used in
this study. The ensemble mean and probability of ECP_PPI positive extremes in the low-top CESM2
model increase from -0.07/5% in P1 to 0.20/11% in P2 and to 0.11/13% in P3, while these values in the
high-top CESM2-WACCM model increase from 0.03/5% in P1 to 0.36/10% in P2 and to 0.27/6% in P3
(Table S5 and Fig. S9 in the Supplement). Both model results are much closer to the changes between
P1 and P2 shown in the reanalysis data than the other CMIP6 models. These increments are also more
significant than the SENSall results of the sensitivity experiment in this study, which might be
attributable to the much stronger climate forcing and fully coupled modeling settings in the CMIP6
simulations.
**4 Discussion and conclusions**
The connection between Arctic sea ice decline and winter air stagnation has been re-examined as a
cause of pollution extremes in China, and the mechanisms for the teleconnection have been explored.
We identified a tropospheric pathway linking the remote sea ice changes in the Arctic to regional
circulation and ventilation responses in eastern China based on statistical analysis and diagnosis of
atmospheric dynamics using the NCEP reanalysis and climate model sensitivity simulation data. The
teleconnection mechanism's evolution can be summarized as follows: In autumn and early winter of
recent years, substantial declines in sea ice in most regions of the Arctic significantly increased upward
heat fluxes during the same period. These changes in surface boundary conditions, especially those that
occurred in the Pacific sector of the Arctic (the East Siberian and Chukchi Seas), induced atmospheric
responses during the following winter producing strengthened eddy kinetic energy fluxes over the mid-
latitudes from the northeastern Pacific to North Africa and strong convergence of anomalous transient
eddy vorticity fluxes in the vicinity of Scandinavia. This transient eddy forcing led to positive
geopotential height tendencies as well as an anomalous ridge in this region throughout the troposphere.
Constructive interference between eddy-induced wave packets and background flow enhanced wave
train propagation across Eurasia, resembling the negative phase of the EU pattern. The high-pressure
anomalies over eastern Asia in the middle and upper troposphere of this teleconnection pattern finally
weakened boundary-layer air ventilation and exacerbated air stagnation extremes in eastern China by
suppressing monsoon northwesterlies and enhancing near surface temperature inversions in this region.
Such meteorological conditions are favorable to air pollutant accumulation and secondary formation.
The occurrence of these teleconnection processes depends on complex interactions between climate
disturbances and its internal variability, which are reflected by diverse climate sensitivity responses in
the full statistical distributions of circulation and ventilation variables. The largest increase of both the
probability (by 132%) and the intensity (by 30%) of monthly air stagnation extremes is found in the
experiment driven by sea ice perturbations over the Pacific sector of the Arctic (the East Siberian and
Chukchi Seas). We emphasize the importance of a full-distribution evaluation, especially for climate
extreme assessment and attribution, considering vastly distinct responses between mean conditions and
extremes and the tendency of underestimated impacts of climate extremes (Schewe et al., 2019). We
note some relevant issues to our analysis that deserve more attention and further investigations.
Firstly, Screen and Simmonds (2014) examined the regional impact of planetary wave changes on
mid-latitude weather extremes and found distinct relationships between quasi-stationary planetary wave
anomalies and regional weather extremes in terms of temperature and precipitation. Specifically, they
found attenuated (amplified) planetary-wave amplitudes accompanying positive temperature extremes
(near-average temperature) over eastern Asia that is different from other regions like central Asia and
western North America (Screen and Simmonds, 2014). This region-dependent relationship can be
attributed to the interaction between anomalous planetary waves and climatological waves that show
quasi-stationary phases. The EU-like teleconnection pattern tends to amplify climatological planetary
waves in the upstream regions such as the North Atlantic and Europe but attenuate climatological waves
in the downstream regions over central and eastern Asia, leading to regional dynamic and
thermodynamic responses as demonstrated in this study. It's worth noting that atmospheric
teleconnection patterns like EU with smaller wave numbers might also be excited by quasi-stationary
spatially inhomogeneous diabatic sources/sinks and orography other than the thermal forcing associated
with the SIC/SST perturbation used in this study. Therefore, we have mainly focused on the relative
changes of teleconnection occurrence probability between the CTRL and sensitivity experiments to
isolate the contributions from this single forcing in the simulated atmospheric system.
Secondly, the climate impacts may vary in response to differing location and magnitude of climate
forcing as we found in our regional sensitivity experiments and this issue is often discussed in other
modeling studies (Screen, 2017; Sun et al., 2015; McKenna et al., 2018). The different responses might
be attributable to different physical mechanisms and atmospheric processes associated with specific
forcing-response relationships. Previous studies proposed multiple pathways of Arctic sea ice impacts
on middle-latitude atmospheric circulation through troposphere-stratosphere coupling and/or
tropospheric processes only (Overland et al., 2016). It is an intriguing question to quantify the relative

importance of different pathways in different case studies. Screen (2017) proposed that a stratospheric pathway dominated the atmosphere responses to sea ice loss in the Barents-Kara Seas whereas tropospheric processes governed wave train responses to sea ice loss in other regions, which is partly consistent with what we found in this study. Similarly, McKenna et al. (2018) also found opposite effects of the regional sea ice forcing on the stratospheric polar vortex in their full-magnitude and half-magnitude forcing experiments, but the tropospheric responses were different between the two experiments with different forcing magnitudes. They suggested that tropospheric processes become more important than stratospheric pathways as the sea ice loss magnitude increases (McKenna et al., 2018). Our modeling results indicate that the tropospheric processes are the key to understand the forcing-response relationship of interest. However, we cannot rule out the possible role of stratospheric changes in midlatitude weather extreme events through stratosphere-troposphere coupling processes (Zhang et al., 2018), and the CESM2 model's sensitivity appears to be stronger when the stratosphere is reasonably resolved in its high-top WACCM version. Multiple dynamic processes and teleconnection pathways associated with different forcing source regions increase the detection difficulty in the whole Arctic perturbation experiment. More detailed sensitivity experiments need to be designed and conducted to evaluate such pathway-dependent effects of Arctic sea ice loss on regional circulation and pollution conditions.

Thirdly, climate responses to Arctic sea ice forcing may also vary on intra-seasonal scales. In a recent study, Lu et al. (2019) revealed an important role of the autumn Arctic sea ice in the phase reversal of the Siberian high in November and December-January. They suggested that the autumn Arctic sea ice loss, especially in the Barents Sea, could induce anomalous upward (downward) surface turbulent heat fluxes in November (December-January). This would strengthen (weaken) the development of the storm track in northeastern Europe and decrease (increase) Ural blockings with accelerated (decelerated) westerlies. With inhibited (enhanced) cold air transport from the Arctic to the Siberian area, a weaker (stronger) Siberian high in November (December-January) would occur thereafter. In our modeling results, we also found significant intra-seasonal variations in simulated atmosphere responses. Figure S10 in the Supplement shows weekly evolution of geopotential height tendencies and anomalies in SENSr2 from late November to February. The negative phases of the EU pattern are more prominent in early winter than in late winter. Better understanding of such intra-seasonal variations could benefit seasonal and sub-seasonal forecasts of regional ventilation and pollution potentials.

Last but not the least, the concurrence of multiple climate drivers and their synergistic climate impacts should be considered. Since many other climate factors such as Eurasian snow cover, ENSO, and PDO also show considerable influence on regional circulation and air pollution in China (Chang et al., 2016; Sun et al., 2018; Zhao et al., 2016; Zhao et al., 2018; Zhang et al., 2019; Zou et al., 2017), more studies with concurrent climate drivers could be conducted to obtain a more comprehensive understanding of climate change impacts. However, these climate drivers may interact with each other in either synergistic or antagonistic ways (Li et al., 2019). Fully sea ice-ocean-atmosphere coupling also allows more interactive dynamic and thermodynamic feedbacks with expanded and enhanced climate responses as suggested by previous studies (Deser et al., 2015; Smith et al., 2017) and by the CMIP6 fully coupled projections in this study. It's a great challenge to distinguish robust and significant responses to climate change from atmospheric internal variability due to relatively low signal-to-noise

ratios (Barnes and Screen, 2014). Callahan et al. (2019) estimated consistent signal-to-noise ratios less than one in multiple regional air stagnation indices for Beijing based on the CESM-LE historical simulations, demonstrating the dominant role of natural variability rather than anthropogenic forcing in modulating regional circulation and ventilation. Divergent consensuses on the climate impact of Arctic amplification on midlatitude severe weather remain an open question for the whole climate science community (Cohen et al., 2020). Therefore, long-term climate simulations with larger ensemble sizes should be conducted to achieve more robust modeling-based findings (Screen and Blackport, 2019). Furthermore, the Arctic sea ice cover reached its historical minimum in the fall of 2012 (Fig. 7a; NSIDC/NASA, 2019). The slowdown of Arctic sea ice loss since then may reflect regional climate internal variability and may have weakened the effect of the Arctic sea ice loss on winter extreme haze occurrence in China in recent years.

## Code and data availability

The NCEP Reanalysis data is provided by the NOAA/OAR/ESRL PSD, Boulder, Colorado, USA, from their web site at https://www.esrl.noaa.gov/psd/data/gridded/data.ncep.reanalysis.html (last access: 18 July, 2019). The MEIC-MIX emission inventory data is available at http://www.meicmodel.org/dataset-mix (last access: 18 July, 2019). The CMIP6 model outputs are distributed by the Earth System Grid Federation (ESGF) at https://esgf-node.llnl.gov/search/cmip6/ (last access: 24 October, 2019). All the CESM-WACCM modeling input and output data are archived on the GLADE and HPSS file systems managed by the Computational & Information Systems Lab (CISL) of NCAR. The simulation results of the control and sensitivity experiments used for the analysis in the main text are deposited at the Figshare website (https://doi.org/10.6084/m9.figshare.11894439). The modeling source code and data materials are available upon request, which should be addressed to Yufei Zou (yufei.zou@pnnl.gov).

## Author contributions

YZ and YW conceived the research idea and designed the climate sensitivity experiments. YZ conducted the modeling experiments and analyzed modeling results with ZX. YZ prepared all the figures and wrote the draft manuscript. All authors discussed the results and revised the manuscript.

## Competing interests

The authors declare no conflict of interest.

## Acknowledgements

We acknowledge high-performance computing support from Yellowstone (CISL, 2016; ark:/85065/d7wd3xhc) and Cheyenne (CISL, 2019; doi:10.5065/D6RX99HX) provided by NCAR's

CISL, sponsored by the National Science Foundation. We thank the Physical Sciences Division (PSD) at the NOAA Earth System Research Laboratory (ESRL) for providing the NCEP/NCAR Reanalysis data. We thank the MEIC team for providing the MEIC-MIX emission inventory data. We acknowledge the World Climate Research Programme, which, through its Working Group on Coupled Modelling, coordinated and promoted CMIP6. We thank the climate modeling groups for producing and making available their model output, the Earth System Grid Federation (ESGF) for archiving the data and providing access, and the multiple funding agencies who support CMIP6 and ESGF. We thank DOE's RGMA program area, the Data Management program, and NERSC for making this coordinated CMIP6 analysis activity possible.

We are also thankful to Robert Black, Yi Deng, Jian Lu, and two anonymous reviewers for their helpful discussion to improve the data analysis and presentation quality of this work.

**Financial support**

This research has been supported by the National Science Foundation Atmospheric Chemistry Program. ZX is supported by the National Natural Science Foundation of China (project number: 41861144014). YZ, HW, and PR are supported by the U.S. Department of Energy (DOE) Office of Science Regional and Global Model Analysis (RGMA) Program. The Pacific Northwest National Laboratory (PNNL) is operated for DOE by Battelle Memorial Institute under contract DE-AC05-76RL01830.

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

**Table 1. The modeling settings of the climate sensitivity experiments using CESM-WACCM**

| Experiment | CTRL | SENSall | SENSr1 | SENSr2 | SENSr3 |
|---|---|---|---|---|---|
| Time period | 30 years | 30 years | 30 years | 30 years | 30 years |
| Horizontal resolution | 1.9°×2.5° | 1.9°×2.5° | 1.9°×2.5° | 1.9°×2.5° | 1.9°×2.5° |
| Vertical level | 70 | 70 | 70 | 70 | 70 |
| Atmosphere | WACCM[a] | WACCM | WACCM | WACCM | WACCM |
| Land | CLM4.0 | CLM4.0 | CLM4.0 | CLM4.0 | CLM4.0 |
| Ocean | Climatology[b] | 2012 Arctic SST | 2012 R1 SST[c] | 2012 R2 SST | 2012 R3 SST |
| Sea ice | Climatology[b] | 2012 Arctic SIC | 2012 R1 SIC | 2012 R2 SIC | 2012 R3 SIC |
| China emissions | MEIC-MIX | MEIC-MIX | MEIC-MIX | MEIC-MIX | MEIC-MIX |
| Other emissions | IPCC AR5 | IPCC AR5 | IPCC AR5 | IPCC AR5 | IPCC AR5 |

[a]: using CAM5 physics package and WACCM_MOZART_MAM3 chemistry package;
[b]: 1981-2010 average based on the HadISST SST and SIC data (Rayner et al., 2003);
[c]: see the main text and Fig. 1b for the R1-R3 region definition;

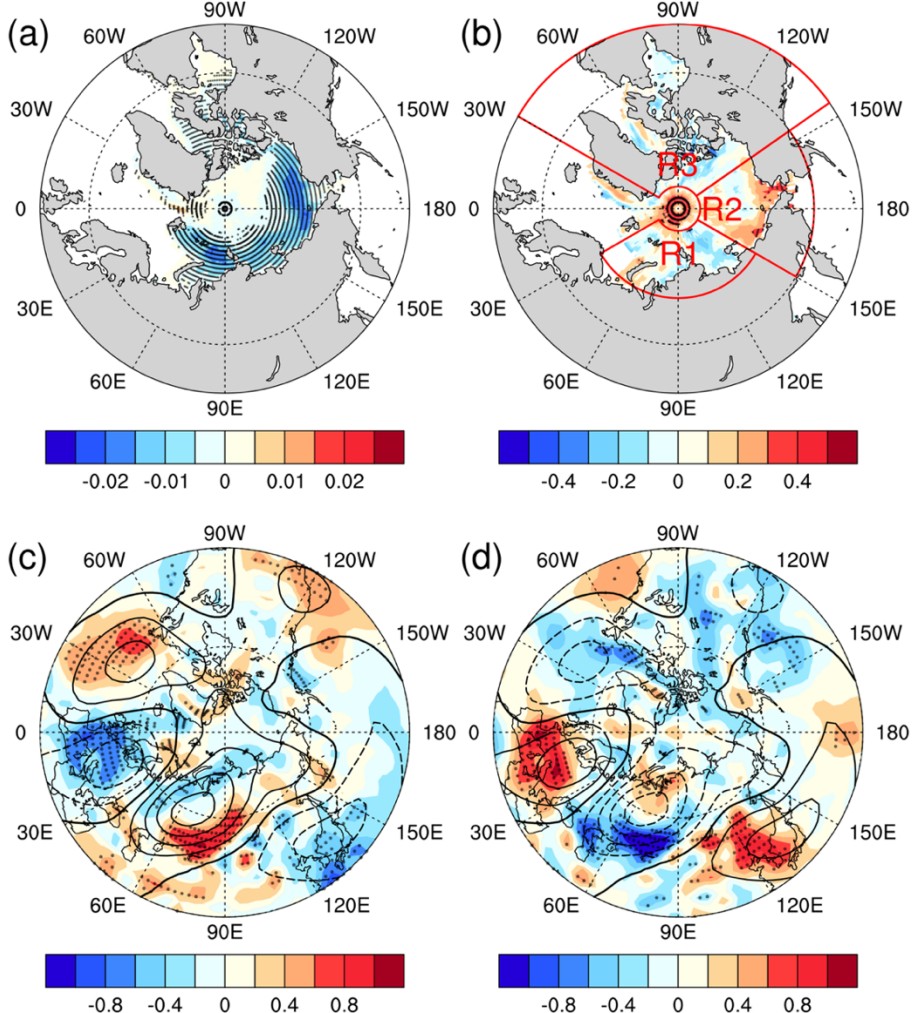

**Figure 1. Relationship between Arctic sea ice changes, EU teleconnection, and pollution ventilation conditions in the Northern Hemisphere based on the reanalysis and observational data. (a) trends of Arctic sea ice changes in autumn and early winter (ASON) of 1980-2017 (color shading in the Arctic region, year$^{-1}$); (b) correlation between the winter EU index and preceding Arctic sea ice concentrations (color shading in the Arctic region, unitless); R1-3 denote the perturbation regions in the three region-specific sensitivity experiments; (c) PPI spatial distributions (color shading, unitless) during the positive phase of EU (contours with interval of 20 m; dashed/solid lines indicate negative/positive geopotential heights at 500 hPa); (d) PPI spatial distributions (color shading, unitless) during the negative phase of EU (contours with interval of 20 m; dashed/solid lines indicate negative/positive geopotential heights at 500 hPa). The stipples over color shading denote the 0.05 significance level based on the two-tailed Student's t-test.**

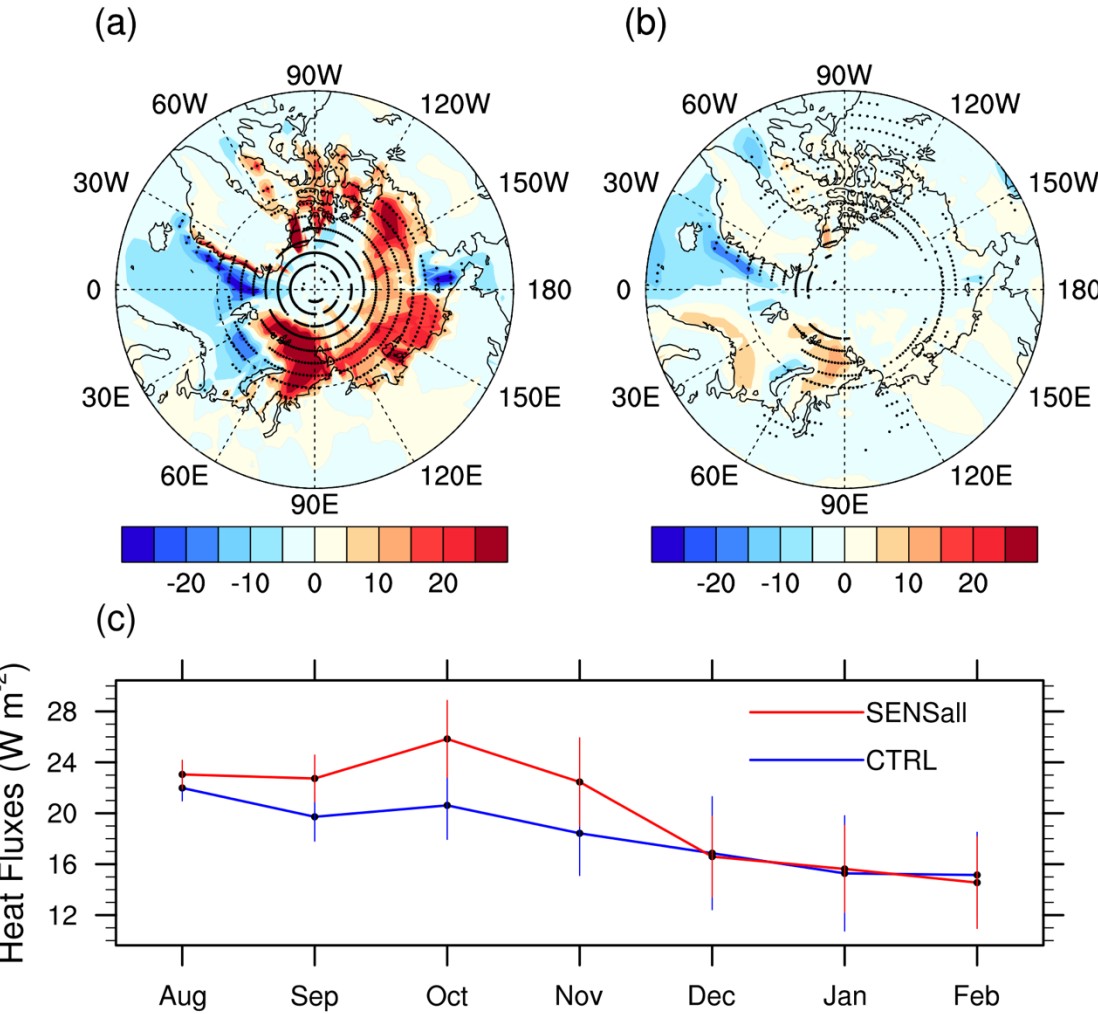

Figure 2. Surface heat flux changes over the Arctic in the WACCM SENSall simulation. (a) differences of surface sensible plus latent heat fluxes (positive upward) between SENSall and CTRL during Aug-Nov; (b) differences of surface sensible plus latent heat fluxes between SENSall and CTRL during Dec-Feb; (c) comparison of regional averaged surface heat fluxes over the Arctic (north of 66.6° N) from August to February. The stipples in (a)/(b) denote the 0.05 significance level. The error bars in (c) denote one standard deviation of the 30-member ensembles in CTRL and SENSall, respectively.

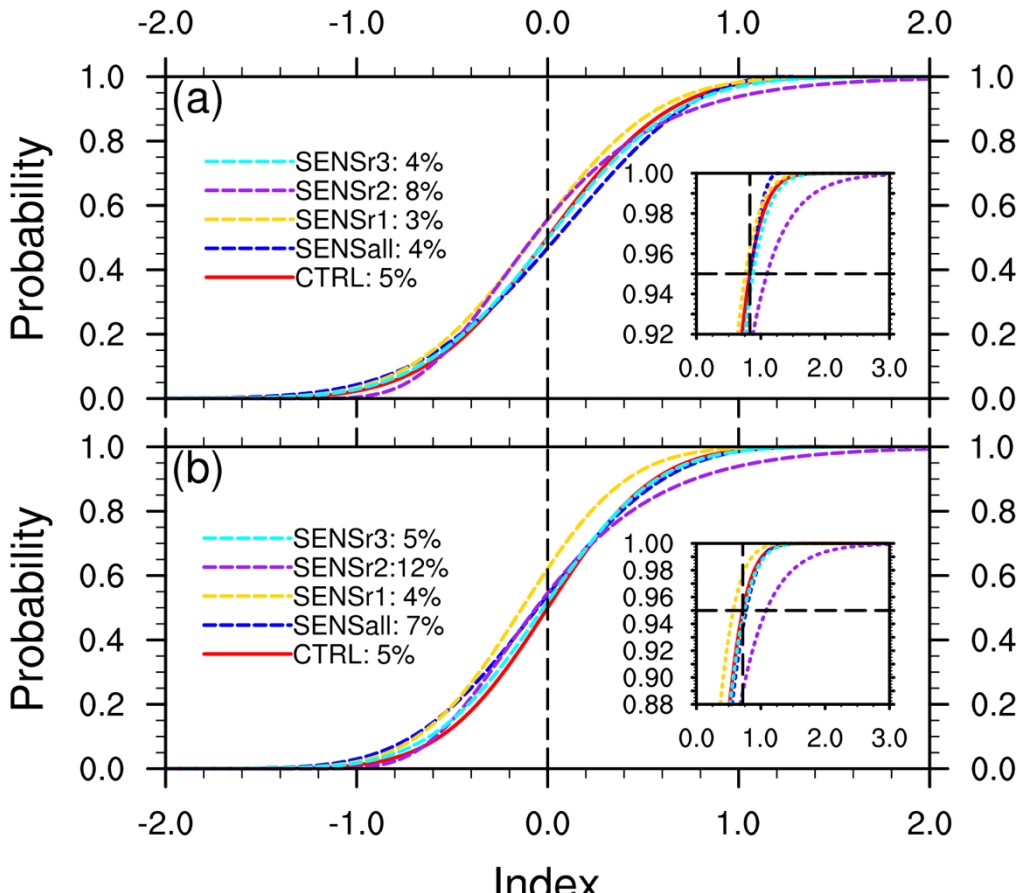

Figure 3. Comparison of the statistical distributions of atmospheric circulation and regional air stagnation indices in the WACCM climate sensitivity experiments. (a) comparison of cumulative distribution functions (CDFs) of the MCA_Z500 index in winter months (Dec, Jan, and Feb). The percentages in the legend are the occurrence probabilities of positive extreme members based on the bootstrap estimation in Table S3 in the Supplement; the inset shows the zoomed-in distributions of positive MCA_Z500 extremes ($\geq MCA\_Z500_{CTRL}^{95th}$) and the black dashed lines in the inset denote the positive extreme threshold ($MCA\_Z500_{CTRL}^{95th} = 0.83$); (b) same as (a) but for the regional averaged ECP_PPI index; the inset shows the zoomed-in distributions of positive PPI extremes ($\geq ECP\_PPI_{CTRL}^{95th}$) and the black dashed lines in the inset denote the positive extreme threshold ($ECP\_PPI_{CTRL}^{95th} = 0.72$).

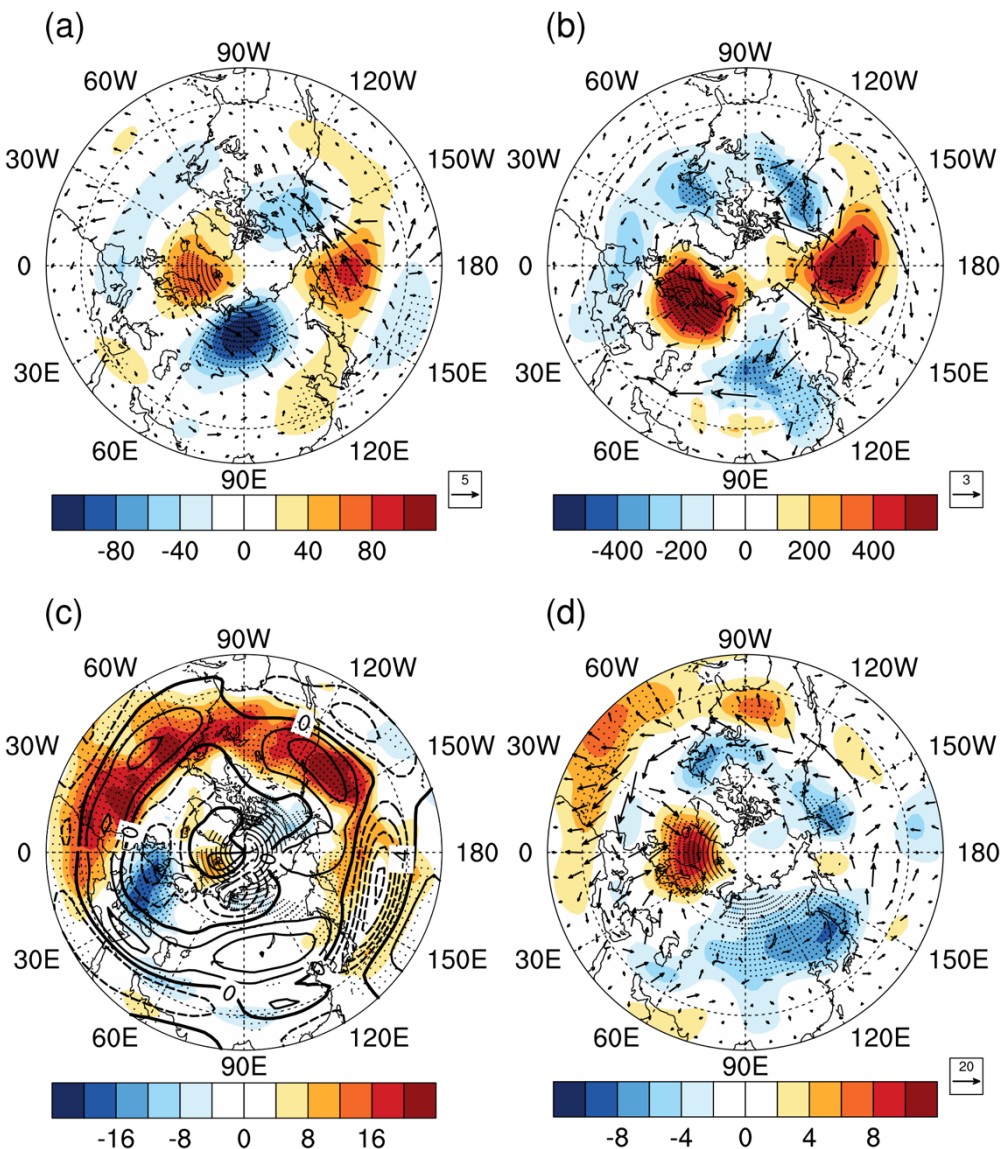

**Figure 4. Atmospheric anomalies in WACCM SENSr2 extreme members with respect to the**
**CTRL ensemble mean. (a) geopotential height (color shading, m) and wave activity flux (vectors,**
**m² s⁻²) anomalies at 250 hPa; (b) sea level pressure (color shading, Pa) and surface wind**
**circulation (vectors, m s⁻¹) anomalies; (c) Anomalous transient eddy kinetic energy (color shading,**
**m² s⁻²) and zonal wind (contours, m s⁻¹) anomalies at 250 hPa; (d) Anomalous E vectors (vectors,**
**m² s⁻²) and transient eddy-induced geopotential height tendencies ($Z_t^{V+H}$) (color shading, m day⁻¹)**
**at 250 hPa. The stipples denote the 0.05 significance level.**

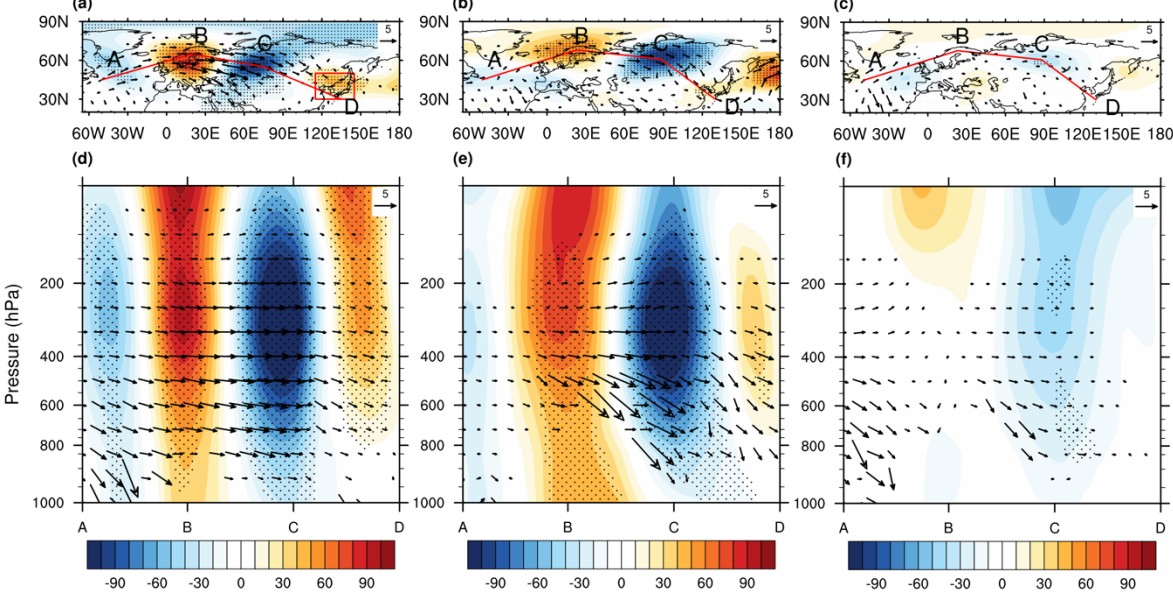

**Figure 5. Comparison of atmospheric anomalies in the NCEP reanalysis data and WACCM experiments. (a) reanalysis-based ensemble mean geopotential heights at 500 hPa (color shading, m) and wave activity flux (WAF) at 250 hPa (vectors, m² s⁻²) of the 30 strongest negative EU months in winter (DJF) of 1951-2019 (relative to the 1981-2010 climatology); (b) same as (a) but based on the SENSr2 extreme members (relative to the CTRL ensemble mean); (c) same as (b) but based on the CTRL counterparts of the SENSr2 extreme members (relative to the CTRL ensemble mean); (d) reanalysis-based vertical cross section of geopotential heights (color shading, m) and WAF (vectors, m² s⁻²) of the ensemble mean negative EU months (relative to the 1981-2010 climatology) along the wave propagation path shown in (a); (e) same as (d) but based on the SENSr2 extreme members (relative to the CTRL ensemble mean); (f) same as (e) but based on the CTRL counterparts of the SENSr2 extreme members (relative to the CTRL ensemble mean). Note that the vertical components of WAF in (c)-(d) were scaled up by 200 for clear illustration. The stipples denote the 0.05 significance level.**

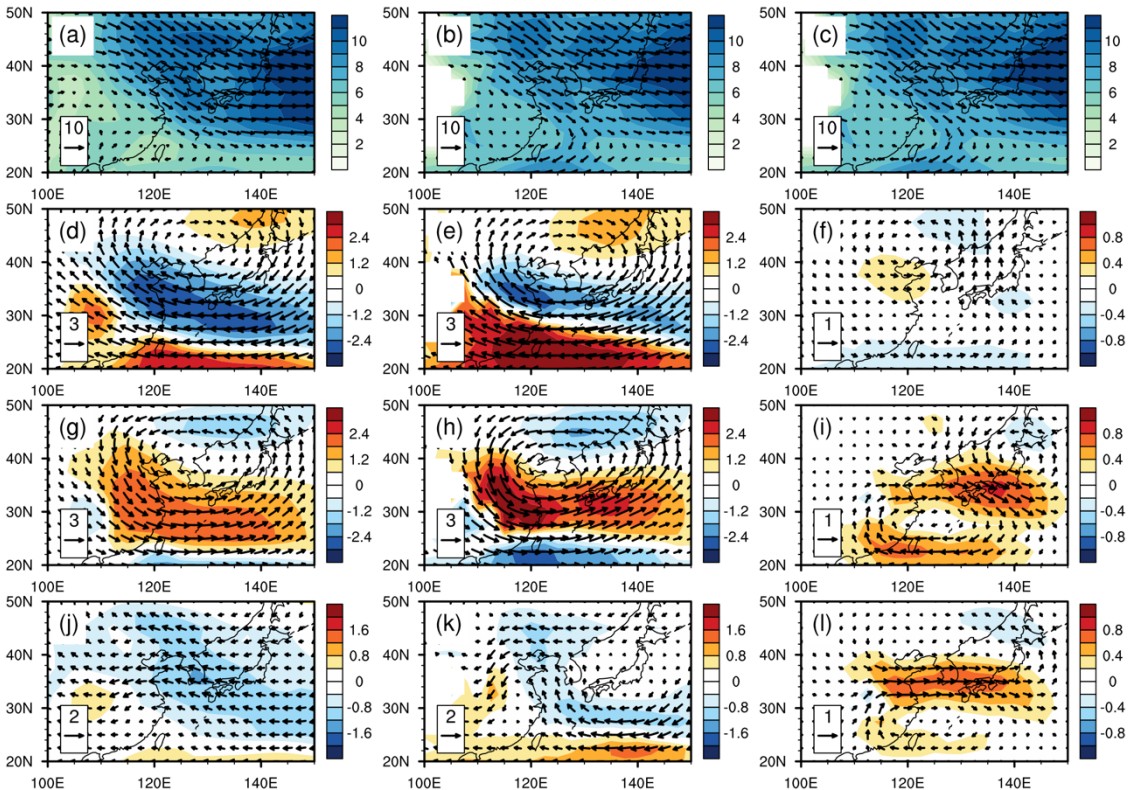

Figure 6. Comparison of winter circulation in East Asia based on the piecewise PV inversion analysis with the NCEP reanalysis and WACCM modeling data. (a) climatological wind speed (color shading, m) and directions (vector, m) at 850 hPa based on the reanalysis data; (b)-(c) same as (a) but based on the WACCM CTRL ensemble mean; (d) reanalysis-based wind circulation change at 850 hPa induced by East Asia PV anomalies (see the red box in Fig. 5a) in the middle to upper troposphere (700-100 hPa) during strong negative EU months in winter; (e) model-based wind circulation change at 850 hPa associated with the middle to upper troposphere PV anomalies over East Asia in the SENSr2 extreme members (relative to the CTRL ensemble mean); (f) same as (e) but in the CTRL counterparts of the SENSr2 extreme members (relative to the CTRL ensemble mean); (g) reanalysis-based wind circulation change at 850 hPa induced by East Asia PV anomalies in the lower troposphere (1000-850 hPa) during strong negative EU months in winter; (h) model-based wind circulation change at 850 hPa associated with the lower troposphere PV anomalies in the SENSr2 extreme members (relative to the CTRL ensemble mean); (i) same as (h) but in the CTRL counterparts of the SENSr2 extreme members (relative to the CTRL ensemble mean); (j) reanalysis-based wind circulation change at 850 hPa induced by East Asia PV anomalies in the whole troposphere (1000-100 hPa) during strong negative EU months in winter; (k) model-based wind circulation change at 850 hPa associated with the whole troposphere PV anomalies in the SENSr2 extreme members (relative to the CTRL ensemble mean); (l) same as (k) but in the CTRL counterparts of the SENSr2 extreme members (relative to the CTRL ensemble mean).

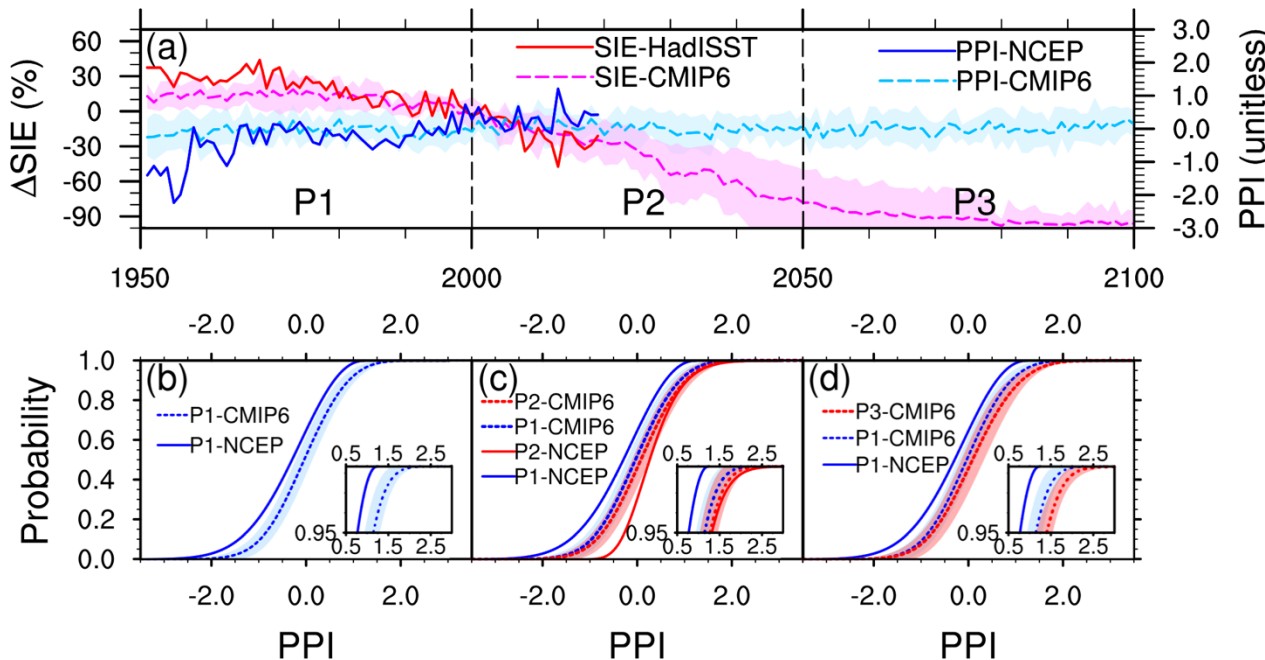

**Figure 7. Historical simulations and future projections (under the SSP5-8.5 scenario) of Arctic sea**
**ice and regional air stagnation in observational and reanalysis data and CMIP6 models. (a) time**
**series of the Arctic SIE relative changes (unit: %; relative to the 1981-2010 climatology) in**
**preceding September and ECP_PPI (unitless) in DJF of the following winter (using years of**
**January for X-axis labeling). The solid lines denote observation- and reanalysis-based Arctic SIE**
**and ECP_PPI from 1950 to 2019. The dashed lines denote ensemble mean and the color shading**
**denotes ±1 standard deviation of the 8 CMIP6 models (see Table S1 for model details) from 1950**
**to 2100. Note that the SIE time series were shifted one year after to be aligned with the ECP_PPI**
**data; (b) comparison of ECP_PPI CDF curves between the NCEP reanalysis data and the CMIP6**
**models in the P1 time period from 1951 to 2000. The inset denotes the distributions of positive**
**extremes (≥ $PPI_{P1}^{95^{th}}$). The color shading denotes ±1 standard deviations in the 8 CMIP6 models;**
**(c) Same as (b) but for the comparison between P1 and P2 (2001-2050) time periods as well as**
**between the NCEP reanalysis data and the CMIP6 models; (d) same as (b) but for the comparison**
**between P1 and P3 (2051-2100) time periods as well as between the NCEP reanalysis data and the**
**CMIP6 models. The model-specific comparison in (b)-(d) are shown in Table S5 and Fig. S9 in the**
**Supplement.**