# Peer review of "Atmospheric teleconnection processes linking winter air stagnation"

_Atmospheric Chemistry and Physics, 2019_

## Referee Comment (RC1) · Anonymous Referee #2 · 3 Jan 2020

GENERAL COMMENTS

This paper uses sensitivity experiments to increase our knowledge and understanding of the dynamics and teleconnections whereby Arctic sea ice decline could influence pollution and air quality in China. This is a timely study with some interesting diagnostics and analysis. The approach toward the sensitivity experiments, the diagnostics defined, and the quality and quantity of figures and tables is all good.

However, I have a couple of concerns as outlined below.

SPECIFIC COMMENTS

[Figure]

I think lines 22 and 32 are not quite correct as they stand. I'd suggest changing "positive correlations with the regional Arctic sea ice decline" to "positive correlations with the regional Arctic sea ice concentrations", and "positive correlation with the declining sea ice" to "positive correlation with sea ice concentrations". My reading of Figure 1 is that the positive correlation between the EU index and sea ice concentrations in R2 (Figure 1b) suggests that the EU index will decrease as sea ice decreases in R2. Then, the positive PPI index in East Asia in the negative phase of the EU (Figure 1d) suggests that this decreasing EU index will lead to increased pollution in East Asia. Therefore, pollution gets worse in East Asia as sea ice declines in R2. Perhaps section 3.1 would be clearer if re-written slightly to emphasize this?

My major concern with this paper is with regards to how relevant this key conclusion is to the real world (which will experience SENSall). Figure 1b shows negative correlations in R1 and R3, so declining sea ice in these regions will improve pollution in East Asia, cancelling the impacts of sea ice decline in R2. Indeed, Tables S3 and S4 indicate no significant difference in PPI between the CTRL and SENSall experiments. This concern, that the substantial conclusions in this paper are all based on SENSr2 which may not be a realistic scenario, will need to be addressed.

Checking of the English grammar and language throughout the manuscript is required. To give a couple of examples: on page 2 line 4 "environmental stressors" should be "environmental stresses", and on page 7 line 21 "are the same with these" should be "are the same as those".

The description of some of the calculations and diagnostics used in the paper could be slightly clearer. For example, in section 2.3, which "statistical functions in Python" were used, and what is meant by "proper distributions"? Also, please expand on the definitions of WSI and ATGI to make clearer exactly how these are calculated.

MINOR COMMENTS AND TYPOGRAPHICAL ERRORS

Page 3, line 32: Person should be Pearson
[Figure]
* * *
Interactive
comment

Page 5: Perhaps change "Fig. 1" to "Fig. 1a" on line 28, and then add reference to "Fig. 1b" on line 31.

Page 10, line 6: "( et al." should be "(et al."

---

## Referee Comment (RC2) · Anonymous Referee #1 · 10 Jan 2020

GENERAL COMMENTS

This paper uses climate model experiments in which regional Arctic sea-ice decline is imposed, combined with analysis of new CMIP6 data, to better understand the dynamical mechanisms by which Arctic sea-ice decline may influence winter haze pollution extremes in China. The main new result reported is that Pacific sector sea-ice loss increases the likelihood and intensity of haze pollution extremes, due to anomalous transient eddy vorticity fluxes amplifying the negative phase of the EU pattern.

Given the substantial impact of haze pollution extremes on public health, this study represents an important contribution to this research area. The variety of methods

used – including targeted single model experiments, new CMIP6 multi-model data, and a variety of interesting diagnostics – is also good. The paper is generally well presented with a good quality and number of figures, and only minor alterations are required to the wording and structure.

However, while this study reports potentially very interesting and impactful results, I am concerned about the statistical robustness of some of the conclusions and, therefore, that the length of the simulations (30 years) may be too short. My specific comments below explain these concerns in detail, which I would like to see addressed.

SPECIFIC COMMENTS

Page 1, line 19: I found the use of the word 'event' a bit confusing in this study, as the extremes analysed are monthly extremes and 'event' – to me anyway – implies a shorter timescale (daily or weekly). It would be helpful to clarify somewhere what is meant by the term 'event' here, or to avoid using the term.

Introduction: This paragraph is far too long, which made the structure of the introduction – which while good – a bit hard to follow. Breaking this up into a few paragraphs would help. The same goes for similarly long paragraphs in other parts of the paper (e.g. page 4 lines 9-40; page 9).

Page 2, lines 32-35: This sentence is a bit misleading, as it implies that there is a scientific consensus that high-latitude climate change influences mid-latitude circulation and weather, when there is not (e.g. https://www.nature.com/articles/s41558-019-0662-y). There is lots of evidence suggesting that Arctic sea-ice loss can have an influence on mid-latitudes, but whether it has in the past or will in the future is more unclear (https://onlinelibrary.wiley.com/doi/full/10.1002/wcc.337). Would be good to rephrase the sentence to reflect this (e.g. 'Given the increasing evidence that climate change – especially that occurring in high-latitude regions – may have an influence on middle-latitude circulation').

Section 2.1: I found this section jumped around a bit in terms of the definitions of the EU pattern and index, the MCA_Z500 pattern, and the PPI. If possible, could this be restructured so that the definition of each is closer to where it is originally introduced?

Page 3, lines 29-30: It would be helpful to properly explain and define the WSI and ATGI.

Page 4, lines 3-5: Do you have a citation for this?

Section 2.2: Are you able to justify using simulations of only 30 years in length? To me this seems rather short, especially considering my comments regarding statistical robustness below. Indeed, Screen et al. 2014 show that the simulated circulation response to sea-ice loss is small compared to internal variability (i.e. there is a low signal-to-noise ratio), and specifically that at least 70 year-long experiments are required to simulate a robust mid-tropospheric response to sea-ice loss (https://link.springer.com/article/10.1007/s00382-013-1830-9). Similarly, simulations submitted to PAMIP (the Polar Amplification Model Intercomparison Project) are required to be at least 100 years long due to this low signal-to-noise ratio (https://www.geosci-model-dev.net/12/1139/2019/). Also, many studies using WACCM to investigate the response to sea-ice loss use longer simulations (e.g. England et al. 2019 use 151 years, https://journals.ametsoc.org/doi/full/10.1175/JCLI-D-17-0666.1; Sun et al. 2015 use 161 years, https://journals.ametsoc.org/doi/full/10.1175/JCLI-D-15-0169.1; Zhang et al. 2018 use 60 years, https://advances.sciencemag.org/content/4/7/eaat6025).

Page 6, lines 4-6: You say that you have 90 samples when conducting this statistical test, and so I presume you are assuming 90 degrees of freedom. However, have you checked whether the MCA_Z500 and/or ECP_PPI indices are autocorrelated (e.g. between consecutive months or lag-1), and therefore whether 90 degrees of freedom is an overestimate?

Page 6, lines 36-39: Relating to the above comment, did you account for autocorrelation when conducting this bootstrapping method (e.g. as done in your previous paper using the moving blocks method https://advances.sciencemag.org/content/3/3/e1602751)? If there is autocorrelation it may be that the uncertainties given by the bootstrap method (Tables S3 and S4) may be underestimated, and therefore the statistical robustness of the differences between the perturbation and CTRL experiments overestimated.

Page 8, lines 21-23: It should be noted that these correlations are not statistically significant at most gridpoints (but perhaps the correlation would be significant if you used an area average?).

Page 9, lines 20-23; Tables S3 and S4: Can you justify why you use the standard deviation here? The numbers in these tables for the SENSr2 experiment contain one of key results of this paper, suggesting that there is an increase in the likelihood and intensity of MCA_Z500 and ECP_PPI positive extremes in response to sea-ice loss in the R2 region. However, by using just the standard deviation it maybe cannot be said that the extremes in SENSr2 are significantly different statistically from those in CTRL. I may be wrong, but a 95% confidence interval seems more appropriate to test whether the difference is statistically robust? Since a 95% confidence interval will be larger, the $9\% \pm 3\%$ figure in Table S3 for SENSr2 MCA_Z500 may not actually be significantly different from CTRL ($5\% \pm 0\%$).

Page 9, line 23 to page 10, line 28: Results relating to changes in the ensemble mean of the MCA_Z500 and ECP_PPI indices are presented and discussed as if they are statistically robust (e.g. 'The differences in the MCA_Z500 and ECP_PPI responses among the four sensitivity experiments in extreme members and ensemble means also suggest complex relationships between Arctic sea ice loss and mid-latitude weather changes'). However, they are only statistically significant for SENSr1 ECP_PPI (p=0.04) – see Table S2. These paragraphs should be edited so that is clear whether the results being presented and discussed are robust or not.

Section 3.4: Why has only the ECP_PPI index been calculated for the CMIP6 results, and not the MCA_Z500 index, when both were for the WACCM results? This seems quite key, since it is MCA_Z500 that demonstrates a dynamical (and therefore more causal) connection between sea-ice loss and ECP_PPI.

Figure 1, Figure S1, Figure 5 (a) and (c): It would be useful to indicate in the captions that these plots are for observational/reanalysis data, rather than for the sensitivity experiments conducted. For Figure 5 (a) and (c) specifically this is mentioned initially, but it would be clearer to say this in the caption after (a) and (c) as well.

Figure 3: In the caption it says 'Atmospheric circulation and regional air stagnation responses to the Arctic sea ice forcing in the WACCM experiments'. However, what is in the figure is the absolute CDFs for the CTRL and SENS experiments, rather than differences between the SENS experiments and CTRL (what is normally defined as the 'response'). The use of 'response' in the caption is therefore confusing and should be changed.

Figure 4: Since these plots show the difference between the SENSr2 extreme members and the CTRL ensemble mean, rather than the CTRL extreme members, these plots do not just show the effect of the sea-ice forcing imposed, but the combined effect of sea-ice loss and internal variability (which causes extreme events without the need for sea-ice loss). The start of the caption ('Winter atmospheric response to the autumn and early winter sea ice change . . .') should therefore be re-phrased. Also - presumably 'winter' means the 'winter mean' here?

Figure 5: Why is there stippling to show statistical significance in all figures except this one?

Figure S4: 'Relative changes' to what?

TECHNICAL CORRECTIONS

Page 3, line 23: Perhaps refer to 'Fig 1 (c) and (d)' instead of just 'Fig 1', since not

referring to whole figure. If there are similar instances in other parts of the paper, could you perhaps change these too for clarity (e.g. page 4, line 1: 'Fig S1 (b)' rather than just 'Fig S1').

Page 3, line 33: I'm not sure the definition of PM_10 would be immediately obvious to all readers, although I could be wrong. Perhaps consider including a very brief definition?

Page 6, line 11: 'these' should be 'those'

Page 9, line 24: 'of two indices' should be 'of the two indices'

Figures 3 and 7: 'inlet' should be 'inset'

Figure 6 (a) and (b): This rainbow colour scale is not colour-blind friendly, so would be hard to interpret for some people. Perhaps use a white to blue scale, with blue indicating stronger winds?

Tables S3 and S4: 'MAC_Z500' in tables should be 'MCA_Z500'

---

## Author Comment (AC1) · 25 Feb 2020

Response to Referee #2:

GENERAL COMMENTS
This paper uses sensitivity experiments to increase our knowledge and understanding of the dynamics and teleconnections whereby Arctic sea ice decline could influence pollution and air quality in China. This is a timely study with some interesting diagnostics and analysis. The approach toward the sensitivity experiments, the diagnostics defined, and the quality and quantity of figures and tables is all good.

However, I have a couple of concerns as outlined below.

Response: Thank you very much for your comments and suggestions. We have revised the manuscript accordingly to address your concerns. Please see below our responses in blue to your specific comments.

SPECIFIC COMMENTS
I think lines 22 and 32 are not quite correct as they stand. I'd suggest changing "positive correlations with the regional Arctic sea ice decline" to "positive correlations with the regional Arctic sea ice concentrations", and "positive correlation with the declining sea ice" to "positive correlation with sea ice concentrations". My reading of Figure 1 is that the positive correlation between the EU index and sea ice concentrations in R2 (Figure 1b) suggests that the EU index will decrease as sea ice decreases in R2. Then, the positive PPI index in East Asia in the negative phase of the EU (Figure 1d) suggests that this decreasing EU index will lead to increased pollution in East Asia. Therefore, pollution gets worse in East Asia as sea ice declines in R2. Perhaps section 3.1 would be clearer if re-written slightly to emphasize this?

Response: Thank you for the suggestion. We revised the sentence here to "The winter EUI shows positive correlations with regional Arctic sea ice concentrations with the strongest correlation over the East Siberian Sea and Chukchi Sea (Fig. 1b), suggesting a decrease of EUI in winter following the sea ice decline over these regions in preceding months.". We also explicitly state in the next paragraph that "Since the EUI shows a positive correlation with declining sea ice in the Pacific sector of the Arctic, we would expect more severe air stagnation over East Asia coinciding with the decrease of EUI and regional Arctic sea ice."
Please see Section 3.1 in the revised manuscript for details.

My major concern with this paper is with regards to how relevant this key conclusion is to the real world (which will experience SENSall). Figure 1b shows negative correlations in R1 and R3, so declining sea ice in these regions will improve pollution in East Asia, cancelling the impacts of sea ice decline in R2. Indeed, Tables S3 and S4 indicate no significant difference in PPI between the CTRL and SENSall experiments. This concern, that the substantial conclusions in this paper are all based on SENSr2 which may not be a realistic scenario, will need to be addressed.

Response:  Thank you for the helpful discussion. We agree with you that the climate system in the real world has much more complex interactive processes among different components than in climate models, and it's still difficult to find a consensus on the potential influence of Arctic warming and sea ice decline on midlatitude extreme weather due to relatively low signal-to-noise ratios (Cohen et al., 2020). Resonating with a common aphorism that "all models are wrong, but some are useful", we interpret our modeling results from the following perspectives:

(1) The key conclusion suggests distinct climate impacts of sea ice loss in different Arctic regions, which is consistent with previous studies (Screen, 2017; Sun et al. 2015, McKenna et al., 2018). Such region-dependent sensitivity along with the associated dynamic process analysis helps to improve understanding of how Arctic warming can affect midlatitude weather and climate extremes. Previously, Barnes and Screen (2015) discussed a similar topic by framing their inquiries around three different questions: "*Can Arctic warming influence the midlatitude jet-stream? (Can it?) Has Arctic warming significantly influenced the midlatitude jet-stream? (Has it?) Will Arctic warming significantly influence the midlatitude jet-stream? (Will it?)*". They concluded that a growing consensus in the model-based studies that Arctic warming *can* significantly influence the midlatitude circulation does not necessarily imply that it *has* in the past, nor that it *will* in the future (Barnes and Screen, 2015). Their thoughts about these three questions provide insight into the discussion here—that is, our modeling results can be more suitably used to answer the "*Can it?*" question rather than the "*Has it?*" one posed in your comment. The answer to the second question is closely related with dynamic processes that actually happened in the real atmosphere, while the answer to the first one is not. Therefore, we can rely on idealized modeling experiments to isolate specific climate effects of interest to answer the "*Can it?*" question, which is the major motive of this study. It's noted that the real world does not experience the SENSall experiment since the sea surface temperatures and sea ice concentrations in the WACCM model experiments are prescribed without the two-way air-sea interactions. These processes are considered by the fully coupled CMIP6 model simulations as shown in Section 3.4 and Fig.7, which also help to answer the "*Can it?*" and "*Will it?*" questions rather than "*Has it?*".

(2) The sensitivity results identify key regions of interest to improve air quality seasonal forecasts. Satellite observations show varying sea ice variability in different years and Arctic regions (Fig. R1/R2). The teleconnection between high latitudes and midlatitudes may be dominated by different dynamic processes associated with changes in subregions of the Arctic from interseasonal to interannual time scales. Previous studies have suggested changing importance of regional sea ice forcing in different months/seasons (Screens, 2017), which is also evident in our SENSall monthly results (Fig. R3/R4). The proposed teleconnection relationship among regional Arctic sea ice, EUI, and ECP_PPI is most prominent in December, with negatively shifted EUI (Fig. R4a) and more positive extremes of ECP_PPI (Fig. R4d) in this month than others. Figure S10 in the Supplement shows similar intra-seasonal variations in SENSr2 atmosphere responses, with more negative EU patterns emerging in early winter than in late winter. However, the studied R2 regional forcing-response relationship can be overwhelmed by other Arctic regional forcings or other climate factors such as tropical forcings in the real atmosphere. That's why a consensus on the climate impact of Arctic warming is difficult to achieve. Our idealized modeling experiments and dynamic diagnosis provide a plausible pathway of how regional Arctic change can affect midlatitude air stagnation weather extremes, while extended and fully coupled experiments are suggested in the discussion part to further investigate the relative importance of different pathways and their roles in the real atmosphere.

[Figure]

Figure R1: Comparison of Arctic sea ice extent time series between 2012 and 2019 (adapted from the NSIDC website at https://nsidc.org/arcticseaicenews/charctic-interactive-sea-ice-graph/; last access: 14 February, 2020).

[Figure]

Figure R2: Comparison of Arctic sea ice extent spatial distributions between 2012/10 and 2019/10 (adapted from the NSIDC website at http://nsidc.org/arcticseaicenews/sea-ice-comparison-tool/; last access: 14 February, 2020)

[Figure]

Figure R3: SENSall geopotential height anomalies at 500 hPa in (a) December; (b) January; (c) February. The stipples denote the 0.05 significance level.

[Figure]

Figure R4: Comparison of KDE-based distribution density estimates in CTRL and SENSall for (a) EUI in December; (b) EUI in January; (c) EUI in February; (d) ECP_PPI in December; (e) ECP_PPI in January; (f) ECP_PPI in February; The dash lines denote ensemble averages.

Checking of the English grammar and language throughout the manuscript is required. To give a couple of examples: on page 2 line 4 "environmental stressors" should be "environmental stresses", and on page 7 line 21 "are the same with these" should be "are the same as those".

Response: We have now checked the English language throughout the manuscript and made a few more corrections.

The description of some of the calculations and diagnostics used in the paper could be slightly clearer. For example, in section 2.3, which "statistical functions in Python" were used, and what is meant by "proper distributions"? Also, please expand on the definitions of WSI and ATGI to make clearer exactly how these are calculated.

Response: Thank you for the suggestion. We have added more detailed descriptions for the calculations and definitions. For example, the Python statistical functions include "normaltest" and "shapiro" for normality tests in Table S2 and Fig. S2/S3, "skew" and "kurtosis" to compute skewness and kurtosis of data sets in Table S2, "norm.fit" to fit normal distributions if the data samples pass normality tests (Fig. S2/S3), or "gumbel_r.fit"/"gumbel_l.fit" to fit right-skewed/left-skewed Gumbel distributions if not (Fig.S2/S3), etc. These explanations are added in Section 2.3 of the revised manuscript.

We also added the descriptions of WSI and ATGI in lines 34-37 of page 3:

"WSI was standardized by subtracting time-averaged climatological mean of near-surface wind speed over the 1981-2010 period from the monthly values at each grid cell and then dividing by its standard deviations in the same period. ATGI was the standardized potential temperature gradient field between 925 and 1000 hPa using the same method. These two indices are used to reflect horizontal and vertical dispersions of near-surface air pollutants, respectively."

We revised the manuscript extensively to address these problems. Please see our revised manuscript with tracked changes for details.

MINOR COMMENTS AND TYPOGRAPHICAL ERRORS
Page 3, line 32: Person should be Pearson.

Response: Thank you. We corrected the typo here.

Page 5: Perhaps change "Fig. 1" to "Fig. 1a" on line 28, and then add reference to "Fig. 1b" on line 31.

Response: Thank you. We added the specific references in the revised manuscript.

Page 10, line 6: "( et al." should be "(et al."

Response: Thank you. It's corrected.

**References**

Barnes, E. A., and Screen, J. A.: The impact of Arctic warming on the midlatitude jet-stream: Can it? Has it? Will it?, WIREs Climate Change, 6, 277-286, 10.1002/wcc.337, 2015.

Cohen, J., Zhang, X., Francis, J. et al.: Divergent consensuses on Arctic amplification influence on midlatitude severe winter weather, Nat. Clim. Chang., https://doi.org/10.1038/s41558-019-0662-y, 10, 20–29, 2020.

Screen, J. A.: Simulated atmospheric response to regional and pan-Arctic sea ice loss, J. Climate, 30, 3945-3962, 2017.

Sun, L., Deser, C., and Tomas, R.A.: Mechanisms of stratospheric and tropospheric circulation response to projected Arctic sea ice loss, J. Climate, 28, 7824-7845, DOI: 10.1175/JCLI-D-15-0169.1, 2015.

McKenna, C. M., Bracegirdle, T. J., Shuckburgh, E. F., Haynes, P. H., Joshi, M. M.: Arctic sea-ice loss in different regions leads to contrasting Northern Hemisphere impacts, Geophys. Res. Lett., https://doi.org/10.1002/2017GL076433, 2018.

---

## Author Comment (AC2) · 25 Feb 2020

Response to Referee #1:

GENERAL COMMENTS
This paper uses climate model experiments in which regional Arctic sea-ice decline is imposed, combined with analysis of new CMIP6 data, to better understand the dynamical mechanisms by which Arctic sea-ice decline may influence winter haze pollution extremes in China. The main new result reported is that Pacific sector sea-ice loss increases the likelihood and intensity of haze pollution extremes, due to anomalous transient eddy vorticity fluxes amplifying the negative phase of the EU pattern.

Given the substantial impact of haze pollution extremes on public health, this study represents an important contribution to this research area. The variety of methods used – including targeted single model experiments, new CMIP6 multi-model data, and a variety of interesting diagnostics – is also good. The paper is generally well presented with a good quality and number of figures, and only minor alterations are required to the wording and structure.

However, while this study reports potentially very interesting and impactful results, I am concerned about the statistical robustness of some of the conclusions and, therefore, that the length of the simulations (30 years) may be too short. My specific comments below explain these concerns in detail, which I would like to see addressed.

Response: Thank you very much for the constructive comments and suggestions. We understand your concerns about the robustness of the modeling results in the manuscript. Therefore, we have conducted additional statistical significance tests to demonstrate that these results are robust. We also revised the manuscript to address your other concerns. Please see below our responses (in blue) to your specific comments.

SPECIFIC COMMENTS
Page 1, line 19: I found the use of the word 'event' a bit confusing in this study, as the extremes analysed are monthly extremes and 'event' – to me anyway – implies a shorter timescale (daily or weekly). It would be helpful to clarify somewhere what is meant by the term 'event' here, or to avoid using the term.

Response: Thank you for the suggestion. Since there are many different types of climate extreme events such as cold extremes, heatwaves, droughts, and extreme precipitation, etc., we want to emphasize here that pollution-related air stagnation extremes are the major focus of this study. To avoid possible confusion with time scale-related interpretation, we rephrased the expression here to "monthly air stagnation extremes" and revised all similar expressions throughout the manuscript.

Introduction: This paragraph is far too long, which made the structure of the introduction – which while good – a bit hard to follow. Breaking this up into a few paragraphs would help. The same goes for similarly long paragraphs in other parts of the paper (e.g. page 4 lines 9-40; page 9).

Response: We have followed the suggestion to break those long paragraphs into shorter ones on page 2, page 4, and page 9. Please see the revised manuscript for details.

Page 2, lines 32-35: This sentence is a bit misleading, as it implies that there is a scientific

consensus that high-latitude climate change influences mid-latitude circulation and weather, when there is not (e.g. https://www.nature.com/articles/s41558-019-0662-y). There is lots of evidence suggesting that Arctic sea-ice loss can have an influence on mid-latitudes, but whether it has in the past or will in the future is more unclear (https://onlinelibrary.wiley.com/doi/full/10.1002/wcc.337). Would be good to rephrase the sentence to reflect this (e.g. 'Given the increasing evidence that climate change – especially that occurring in high-latitude regions – may have an influence on middlelatitude circulation').

Response: We agree that there are lots of discussion and ongoing debates on this topic. Knowledge gaps regarding complex interactions between high-latitude and mid-latitudes and physical pathways behind these phenomena still exist. A few climate modeling studies have been conducted to narrow down the uncertainty associated with the influence of high-latitude climate change on mid-latitude weather extremes. Our study was motivated and inspired by these discussions and modeling efforts. To clarify on the current research status, we have rephrased the text as suggested and added more specific discussion and references in lines 2-5 of page 3: "Several possible dynamic pathways linking Arctic warming to midlatitude weather extremes have been proposed and investigated in the past few years (Barnes and Screen, 2015; Overland et al., 2016). However, the observational data and modeling results are sometimes contradictory and are open to different interpretations (Cohen et al., 2020)".

Section 2.1: I found this section jumped around a bit in terms of the definitions of the EU pattern and index, the MCA_Z500 pattern, and the PPI. If possible, could this be restructured so that the definition of each is closer to where it is originally introduced?

Response: Thank you for the suggestion. We revised this section to more clearly describe all the indices used in the manuscript. Please see Section 2.1 in the revised manuscript for details.

Page 3, lines 29-30: It would be helpful to properly explain and define the WSI and ATGI.

Response: The two indices are defined and explained in lines 34-37 of page 3: "WSI was standardized by subtracting time-averaged climatological mean of near-surface wind speed over the 1981-2010 period from the monthly values at each grid cell and then dividing by its standard deviations in the same period. ATGI was the standardized potential temperature gradient field between 925 and 1000 hPa using the same method. These two indices are used to reflect horizontal and vertical dispersions of near-surface air pollutants, respectively."

Page 4, lines 3-5: Do you have a citation for this?

Response: This statement is based on the similarity between MCA_Z500 and EU as well as other teleconnection patterns such as the East Atlantic (EA) pattern (https://www.cpc.ncep.noaa.gov/data/teledoc/ea_map.shtml) and the East Atlantic/Western Russia (EA/WR) pattern (https://www.cpc.ncep.noaa.gov/data/teledoc/eawruss.shtml) over East Asia in winter (e.g., January patterns), as shown in Fig. R1.

[Figure]

Figure R1: 500 hPa geopotential height anomalies (unit: m) of the East Atlantic pattern (left) and the East Atlantic/Western Russia pattern (right) in different months. These plots are adapted from the NOAA Climate Prediction Center (CPC) website (https://www.cpc.ncep.noaa.gov/data/teledoc/telecontents.shtml; last access: 14 February 2020).

The major difference between EU and other planetary-scale teleconnection patterns is in the wave propagation pathways in the upstream regions such as the Atlantic and Europe, while they share similar configurations in the downstream regions over East Asia. All these teleconnection patterns can be excited by either internal variability or localized forcings (Simmons et al., 1983; Sardeshmukh et al., 1988; Liu et al., 2014; Lim, 2015). To clarify this, we added examples and references in lines 15-18 of page 4 in the revised manuscript as:
"However, it's worth noting that this regional MCA_Z500 pattern can also be excited by other large-scale teleconnection processes such as the East Atlantic pattern or the East Atlantic/Western Russia pattern associated with both natural variability and perturbed Rossby wave activity (Lim, 2015; Simmons et al., 1983)."

Section 2.2: Are you able to justify using simulations of only 30 years in length? To me this seems rather short, especially considering my comments regarding statistical robustness below. Indeed, Screen et al. 2014 show that the simulated circulation response to sea-ice loss is small compared to internal variability (i.e. there is a low signal-to-noise ratio), and specifically that at least 70 year-long experiments are required to simulate a robust mid-tropospheric response to sea-ice loss (https://link.springer.com/article/10.1007/s00382-013-1830-9). Similarly, simulations submitted to PAMIP (the Polar Amplification Model Intercomparison Project) are required to be at least 100 years long due to this low signal-to-noise ratio (https://www.geosci-model-dev.net/12/1139/2019/). Also, many studies using WACCM to investigate the response to sea-ice loss use longer simulations (e.g. England et al. 2019 use 151 years, https://journals.ametsoc.org/doi/full/10.1175/JCLI-D-17-0666.1; Sun et al. 2015 use 161 years,

https://journals.ametsoc.org/doi/full/10.1175/JCLI-D-15-0169.1; Zhang et al. 2018 use 60 years, https://advances.sciencemag.org/content/4/7/eaat6025).

Response: Thank you for the comment and references. Several studies, including those in your comment, have indicated that the signal-to-noise ratio associated with the Arctic influence on midlatitude weather is lower than internal variability, which motivated the long-term simulations in those studies to try to isolate a robust atmospheric response in the middle latitudes to Arctic sea ice loss and Arctic amplification. However, most, if not all, Arctic-midlatitude impact studies focused on the response in ensemble seasonal mean state rather than monthly extreme values in our case. We want to emphasize that the modeling responses could be very different in terms of these two metrics. This is evident by comparing the changes in ensemble mean values (Table S2 in the Supplement) with those in extreme values (Table S3/S4 in the Supplement) of each sensitivity experiment. It can also be clearly demonstrated by the following conceptual changes in temperature distribution and their effects on extreme values (Fig. R2). In this IPCC report (2012), three distinct distribution changes in response to climate change have been proposed: shifted mean, increased variability, and changed symmetry, which suggest complex relationship between changes in ensemble mean and extreme values. We followed this analysis framework to characterize modeling responses in our climate sensitivity experiments and found the SENSr2 results of interest fall into the "Changed Symmetry" category (as shown in Fig. S2/S3 in the Supplement).

[Figure]

[Figure]

[Figure]

Figure R2: The effect of distribution changes on temperature extremes. Different changes in temperature distributions between present and future climate and their effects on extreme values of the distributions: (a) effects of a simple shift of the entire distribution toward a warmer climate; (b) effects of an increase in temperature variability with no shift in the mean; (c) effects of an altered shape of the distribution, in this example a change in asymmetry toward the hotter part of the distribution. This plot is adopted from Figure SPM. 3 in IPCC (2012).

To evaluate statistical significance of the changes in positive extreme probability, we repeatedly resample a subset of modeling years in SENSr2 for 10,000 times and then use a non-parametric kernel density estimation (KDE) function to re-estimate the probability of ECP_PPI (Fig. R3) positive extremes in each subset comparing with their CTRL counterpart. We try two different methods of resampling: without replacement and with replacement for multiple subset sizes (10, 15, 20, 25, 30). Sampling without replacement does not allow duplicated modeling years while sampling with replacement generates much more combinations and larger variances of subsets. Since there are numerous combinations of resampled subsets (except the subset size of 30 years without replacement, which has only one unique combination of all data), we plot the empirical probability density distributions of positive extreme probabilities using kernel density estimation for each subset size (similar to Fig. 8 in Screen et al., 2014). Please note that the actual sample size in each subset should be multiplied by 3 because we use monthly data in winter (Dec-Jan-Feb) rather than seasonal average to detect climate extremes. For example, the total sampling size for the subset of 10 years is 3 months × 10 years = 30 months. The autocorrelation among these winter months is low and insignificant (please see the response to the next question for details).

After obtaining these empirical PDFs, we estimate the corresponding value of the CTRL positive extreme probability in these PDFs to test the hypothesis that the SENSr2 positive extreme probability is significantly larger than the CTRL run (the CTRL positive extreme probability is always 0.05 since the 95th percentile of CTRL data is chosen as the positive extreme threshold). As shown in Fig. R3 below, the chance of SENSr2 ECP_PPI positive extreme probability $\leq$ CTRL ECP_PPI positive extreme probability (0.05) is about 3% when the subset size exceeds 15 years (45 months) without replacement (Fig. R3a), or when the subset size exceeds 25 years (75 months) with replacement (Fig. R3b). Another way to demonstrate this is to plot the positive extreme probability estimates and their 95% percentile ranges against different ensemble sizes (Fig. R4; we updated Fig. S3/S4 in the supplement using the same method here), which suggest the same conclusion. The ensemble averaged estimates of the SENSr2 ECP_PPI positive extreme probability are also quite similar among different ensemble sizes (~0.11) and more than double of the CTRL positive extreme probability (0.05). Therefore, we are confident that the current modeling simulation length of 30 years is long enough to detect significant extreme probability changes, which is the primary research objective of this study.

[Figure]

Figure R3: KDE-based probability estimates of ECP_PPI positive extremes in SENSr2 based on different ensemble sizes of subsets (a) without and (b) with replacement in bootstrap resampling (n=10,000). The p values on bottom-right corners are the probabilities of 0.05 (the CTRL positive extreme probability shown as the black dash line) in each PDF curve. The colored dash lines are ensemble averaged probabilities of SENSr2 positive extremes for each subset size. Note that no PDF curve is available for nsize=30 without replacement in (a) because of the uniqueness of the sampling combination.

[Figure]

Figure R4: Comparison of KDE-based probability estimates of ECP_PPI positive extremes based on different ensemble sizes of subsets (a) without and (b) with replacement in bootstrap resampling (n=10,000). The error bars denote the 95% percentile range (2.5% to 97.5%) for positive extreme probability values at each ensemble size.

Page 6, lines 4-6: You say that you have 90 samples when conducting this statistical test, and so I presume you are assuming 90 degrees of freedom. However, have you checked whether the MCA_Z500 and/or ECP_PPI indices are autocorrelated (e.g. between consecutive months or lag-1), and therefore whether 90 degrees of freedom is an overestimate?

Response: Thank you for the suggestion. We can treat each sequence of monthly data in Dec, Jan, and Feb as three sampling groups. The essence of this question is whether two consecutive groups of monthly data are independent or not. We test the lag-1 relationship in both MCA_Z500 and ECP_PPI indices by checking the Pearson correlation coefficients between two consecutive monthly groups. If they are not independent from each other, then we would expect statistically significant correlations between these paired groups. Table R1 and Table R2 show the correlation coefficients for both indices and their corresponding two-tailed p-values,

respectively, suggesting insignificant correlations in most MCA_Z500 pairs and all ECP_PPI pairs.

Table R1: Correlation coefficients of the MCA_Z500 and ECP_PPI indices between two consecutive months in each modeling experiment

| r | MCA_Z500 | | ECP_PPI | |
|---|---|---|---|---|
| | Dec-Jan | Jan-Feb | Dec-Jan | Jan-Feb |
| CTRL | 0.27 | 0.01 | -0.02 | -0.28 |
| SENSall | 0.43 | 0.05 | 0.24 | 0.19 |
| SENSr1 | 0.003 | 0.28 | -0.04 | -0.03 |
| SENSr2 | 0.17 | 0.53 | 0.07 | -0.17 |
| SENSr3 | 0.25 | 0.03 | 0.14 | -0.23 |

Table R2: Two-tailed p-value of the MCA_Z500 and ECP_PPI correlation coefficients between two consecutive months in each modeling experiment

| p-value | MCA_Z500 | | ECP_PPI | |
|---|---|---|---|---|
| | Dec-Jan | Jan-Feb | Dec-Jan | Jan-Feb |
| CTRL | 0.15 | 0.95 | 0.90 | 0.14 |
| SENSall | 0.02 | 0.81 | 0.20 | 0.31 |
| SENSr1 | 0.99 | 0.14 | 0.84 | 0.87 |
| SENSr2 | 0.37 | 0.003 | 0.72 | 0.37 |
| SENSr3 | 0.18 | 0.86 | 0.47 | 0.22 |

Since the Pearson correlation coefficient is highly sensitive to outliers, we also plot the scatter plots based on the winter consecutive monthly MCA_Z500 data in SENSall and SENSr2 that show possible correlations. As shown in the plots, the Pearson correlations are mainly contributed by two MCA_Z500 outliers (in the red circle) on bottom-left corners between December and January in SENSall (Fig. R5a), and one MCA_Z500 outlier (in the red circle) on top-right corner between January and February in SENSr2 (Fig. R6b). After removing these outliers, the correlations would largely decrease to insignificant levels (SENSall: (r=0.13, p=0.52); SENSr2: (r=0.36, p=0.06) after removing the outliers in red circles). Another way to show the large impact of outliers on the Pearson correlation coefficients is to use the non-parametric Kendall rank correlation as an alternative, which is more suitable for small sample sizes without the Gaussian distribution assumption. The Kendall rank correlation coefficients for these MCA_Z500 data in Dec-Jan of SENSall and Jan-Feb of SENSr2 are (r=0.16, p=0.23) and (r=0.26, p=0.04), respectively. Both are much smaller than the Pearson ones listed in the above tables. Actually, the lifetime of most severe pollution events is shorter than one month, and the memory effect of the atmosphere is also short. Therefore, we feel it's acceptable to treat these monthly data as independent samples and the degree of freedom of 90 is considered a roughly accurate estimate for the statistical tests in the manuscript.

[Figure]

Figure R5: Scatter plots for the MCA_Z500 and ECP_PPI indices in consecutive months of the SENSall experiment. (a) the paired MCA_Z500 indices in December and January; (b) the paired MCA_Z500 indices in January and February; (c) the paired ECP_PPI indices in December and January; (d) the paired ECP_PPI indices in January and February. The red circle in (a) shows the outliers contribute largely to the Pearson correlation coefficient.

[Figure]

Figure R6: Scatter plots for the MCA_Z500 and ECP_PPI indices in consecutive months of the SENSr2 experiment. (a) the paired MCA_Z500 indices in December and January; (b) the paired MCA_Z500 indices in January and February; (c) the paired ECP_PPI indices in December and January; (d) the paired ECP_PPI indices in January and February. The red circle in (b) shows the outlier contributes largely to the Pearson correlation coefficient.

Page 6, lines 36-39: Relating to the above comment, did you account for autocorrelation when conducting this bootstrapping method (e.g. as done in your previous paper using the moving blocks method https://advances.sciencemag.org/content/3/3/e1602751)? If there is autocorrelation it may be that the uncertainties given by the bootstrap method (Tables S3 and S4) may be underestimated, and therefore the statistical robustness of the differences between the perturbation and CTRL experiments overestimated.

Response: As shown in our response to the previous comment, the autocorrelation in monthly data is negligible in most cases. Therefore, we used the standard bootstrapping method in this manuscript. The reason we used the moving block bootstrap method in our previous study (Zou et al., 2017) is that we used daily data in that study. The autocorrelation problem in these daily time series is much more severe than the monthly data used in this study, so the moving block bootstrap method was applied there. For the monthly data with less concern about autocorrelation, the standard bootstrap method was applied in the previous study (Zou et al., 2017) that is the same with the practice here.

Page 8, lines 21-23: It should be noted that these correlations are not statistically significant at most grid points (but perhaps the correlation would be significant if you used an area average?).

Response: This figure helps to identify the Arctic sub-regions with potential influence on the atmospheric teleconnection as well as regional ventilation in China. If averaging SIC over those R2 areas with positive correlations with the EU index, the regional averaged SIC-EU correlation coefficient is r=0.38 (p=0.02), which is statistically significant at the 0.05 significance level. We added this regional averaged correlation to lines 3-4 of page 9 in the revised manuscript.

Page 9, lines 20-23; Tables S3 and S4: Can you justify why you use the standard deviation here? The numbers in these tables for the SENSr2 experiment contain one of key results of this paper, suggesting that there is an increase in the likelihood and intensity of MCA_Z500 and ECP_PPI positive extremes in response to sea-ice loss in the R2 region. However, by using just the standard deviation it maybe cannot be said that the extremes in SENSr2 are significantly different statistically from those in CTRL.
I may be wrong, but a 95% confidence interval seems more appropriate to test whether the difference is statistically robust? Since a 95% confidence interval will be larger, the 9% 3% figure in Table S3 for SENSr2 MCA_Z500 may not actually be significantly different from CTRL (5% 0%).

Response: Thank you for the suggestion. We redid the bootstrap analysis with replacement for 10,000 times to estimate the 95% percentile range for all the indices listed in Table S3 and Table S4. Please see below the updated tables (we included here for your convenience):

Table S3. The bootstrap (nboot=10000) estimates (ensemble mean and 95% percentile range) of positive extreme probabilities of the MCA_Z500 and ECP_PPI indices in the WACCM experiments

|  | CTRL | SENSall | SENSr1 | SENSr2 | SENSr3 |
|---|---|---|---|---|---|
| MCA_Z500 | 5.0% | 3.7% (0-13.5%) | 3.3% (0-9.2%) | 7.5% (0.8-16.4%) | 4.1% (0-12.8%) |
| ECP_PPI | 5.0% | 7.0% (0.7-16.1%) | 4.1% (0.4-9.2%) | 11.6% (5.2-18.4%) | 5.0% (0.2-11.0%) |

Table S4. The bootstrap (nboot=10000) estimates (ensemble mean and 95% percentile range) of positive extreme intensities of the MCA_Z500 and ECP_PPI indices in the WACCM experiments

|  | CTRL | SENSall | SENSr1 | SENSr2 | SENSr3 |
|---|---|---|---|---|---|
| MCA_Z500 | 1.14 (0.75-1.72) | 1.00 (0.77-1.35) | 1.07 (0.81-1.44) | 1.27 (0.90-1.68) | 1.03 (0.77-1.41) |
| ECP_PPI | 0.86 (0.63-1.40) | 0.91 (0.70-1.25) | 0.94 (0.72-1.31) | 1.12 (0.90-1.42) | 0.84 (0.66-1.13) |

The new estimates don't change our conclusion in the manuscript, suggesting significantly increased probability and intensity of ECP_PPI in SENSr2. This is also evident in Fig. R3b of the previous response. The increase in MCA_Z500 is less significant than that in ECP_PPI, which might be attributed to the smaller signal-to-noise ratio in large-scale dynamic processes. Extended climate sensitivity experiments could be conducted in the future to evaluate the robustness of these large-scale dynamic responses.

Page 9, line 23 to page 10, line 28: Results relating to changes in the ensemble mean of the MCA_Z500 and ECP_PPI indices are presented and discussed as if they are statistically robust (e.g. 'The differences in the MCA_Z500 and ECP_PPI responses among the four sensitivity experiments in extreme members and ensemble means also suggest complex relationships between Arctic sea ice loss and mid-latitude weather changes'). However, they are only statistically significant for SENSr1 ECP_PPI (p=0.04) – see Table S2. These paragraphs should be edited so that is clear whether the results being presented and discussed are robust or not.

Response: We rewrote the paragraphs to clearly indicate the robustness of changes in both ensemble mean and extreme values of both indices. Please see below the updated paragraphs in Section 3.2 of the revised manuscript.

"To examine the regional circulation and ventilation responses to these changes in the high latitudes, we fit the CDF and PDF curves of MCA_Z500 and ECP_PPI based on CTRL and SENS monthly results in winter. Figure 3 shows the CDF changes of simulated MCA_Z500 (Fig. 3a) and ECP_PPI indices (Fig. 3b) between sensitivity and CTRL experiments. It is clear that both indices show more significant changes in their extreme members than in medians or ensemble means, especially in SENSr2 driven by SIC and SST changes in the Pacific sector of the Arctic (R2 in Fig. 1b). In SENSr2, the occurrence probability of MCA_Z500 positive extremes increases by 50% from 5.0 to 7.5% (95th percentile range: 0.8-16.4%) (Fig. 3a; Table S3 in the Supplement), while the ECP_PPI positive extremes increases by 132% to 11.6% (95% percentile range: 5.2-18.4%) (Fig. 3b; Table S3 in the Supplement). Meanwhile, the intensity of positive extreme values of the two indices also increases by 11% and 30%, respectively (Table S4 in the Supplement). The increase in the teleconnection pattern index MCA_Z500 is less significant than that in the regional air stagnation index ECP_PPI, suggesting a potential

nonlinear relationship between large-scale circulation and regional stagnation. Only SENSr2 shows statistically significant increases of ECP_PPI in terms of positive extreme probability and intensity, and the significance of such increases is independent from the fitting method being used (i.e., still valid with nonparametric curve fitting). The substantially increased positive extremes in SENSr2 contribute to the positive responses in its ensemble mean, making SENSr2 the only sensitivity experiment with positive ensemble mean ECP_PPI (0.03, not statistically significant). In comparison, other SENS experiments generally show negative ensemble mean ECP_PPI values due to left-shifted CDF curves at most percentiles. For instance, SENSr1 is the only experiment showing robustly decreased ECP_PPI at all percentiles in its CDF curve (Fig. 3b), contributing to its negative ensemble mean of ECP_PPI (-0.13) that is statistically significant at the 0.05 significance level (Table S2 in the supplement). This result implies an overall improvement of the ECP regional ventilation driven by the SIC and SST changes in the Barents-Kara Seas (R1 in Fig. 1b), while the ventilation responses are more random driven by sea ice loss in other Arctic regions."

Section 3.4: Why has only the ECP_PPI index been calculated for the CMIP6 results, and not the MCA_Z500 index, when both were for the WACCM results? This seems quite key, since it is MCA_Z500 that demonstrates a dynamical (and therefore more causal) connection between sea-ice loss and ECP_PPI.

Response: Thank you for the suggestion. We have now added the time series and changes of MCA_Z500, based on the reanalysis and CMIP6 results, in a new supplementary Figure S8 (shown as Fig. R7 here). The MCA_Z500 projection results also show right-shifted positive extremes in future time periods, with the largest shift emerging during the P3 period in concurrence with the strongest decline of Arctic sea ice. Please see Section 3.4 in the revised manuscript for details.

[Figure]

Figure R7. Historical simulations and future projections (under the SSP5-8.5 scenario) of Arctic sea ice and regional circulation in observational and reanalysis data and CMIP6 models. (a) time

series of the Arctic SIE relative changes (unit: %; relative to 1981-2010) in preceding September and MCA_Z500 (unitless) in DJF of the following winter (using years of January for X-axis labeling). The solid lines denote observation- and reanalysis-based Arctic SIE and MCA_Z500 from 1950 to 2019. The dashed lines denote ensemble mean and the color shading denotes ±1 standard deviation of the 8 CMIP6 models (see Table S1 for model details) from 1950 to 2100. Note that the SIE time series were shifted forward by one year to align with the MCA_Z500 data; (b) comparison of MCA_Z500 CDF curves between the NCEP reanalysis data and the CMIP6 models in the P1 time period from 1951 to 2000. The inset denotes the distributions of positive extremes ($\geq MCA\_Z500_{P1}^{95^{th}}$). The color shading denotes ±1 standard deviations in the 8 CMIP6 models; (c) Same as (b) but for the comparison between P1 and P2 (2001-2050) time periods as well as between the NCEP reanalysis data and the CMIP6 models; (d) same as (b) but for the comparison between P1 and P3 (2051-2100) time periods as well as between the NCEP reanalysis data and the CMIP6 models.

Figure 1, Figure S1, Figure 5 (a) and (c): It would be useful to indicate in the captions that these plots are for observational/reanalysis data, rather than for the sensitivity experiments conducted. For Figure 5 (a) and (c) specifically this is mentioned initially, but it would be clearer to say this in the caption after (a) and (c) as well.

Response: We add the descriptions in the figure captions as suggested.

Figure 3: In the caption it says 'Atmospheric circulation and regional air stagnation responses to the Arctic sea ice forcing in the WACCM experiments'. However, what is in the figure is the absolute CDFs for the CTRL and SENS experiments, rather than differences between the SENS experiments and CTRL (what is normally defined as the 'response'). The use of 'response' in the caption is therefore confusing and should be changed.

Response: We change the description here to "Comparison of the statistical distributions of atmospheric circulation and regional air stagnation indices in the WACCM climate sensitivity experiments" for clarification.

Figure 4: Since these plots show the difference between the SENSr2 extreme members and the CTRL ensemble mean, rather than the CTRL extreme members, these plots do not just show the effect of the sea-ice forcing imposed, but the combined effect of sea-ice loss and internal variability (which causes extreme events without the need for sea-ice loss). The start of the caption ('Winter atmospheric response to the autumn and early winter sea ice change : : :') should therefore be re-phrased. Also - presumably 'winter' means the 'winter mean' here?

Response: Thank you for the suggestion. We rephrase the Fig.4 caption to "Atmospheric anomalies in WACCM SENSr2 extreme members with respect to the CTRL ensemble mean". These extreme members spread in different winter months. Here the anomalies are based on the differences between the average of these extreme members and the CTRL average.
In the dynamic diagnosis part, we attempt to answer the following two questions:
    (1) How does severe air stagnation occur in these SENSr2 extreme members?
    (2) Why are there more and intensified air stagnation extremes in SENSr2?
As indicated by your comments, the extreme weather in these ensemble members could result from interactions between atmospheric internal variability and Arctic sea ice forcing. And we do

find constructive interference between sea ice-related anomalous wave activity and the background flow (Fig. S7 in the supplement). Therefore, we use Fig. 4 in combination with the following figures (Fig. 5/6 in the revised manuscript) to answer the first question, and then use Fig. 5/6 and Fig. S7 in the supplement to answer the second question. Please see Section 3.3 of the revised manuscript for detailed analysis.

Figure 5: Why is there stippling to show statistical significance in all figures except this one?

Response: We didn't add stippling to this figure in the previous version because it already has 3 layers (shading, contour, and vectors). Adding stipples would further increase its complexity. In the revised manuscript, we update Fig. 5 in the manuscript by separating the SENSr2 extreme members from their CTRL counterparts and adding stipples for significance tests in each subplot as suggested (shown as Fig. R8 here). A new figure is also added in the supplement (Fig. S7) to isolate the difference between the SENSr2 extreme members and their CTRL counterparts directly (SENSr2$_{extreme}$-CTRL$_{counterpart}$).

[Figure]

Figure R8: Comparison of atmospheric anomalies in the NCEP reanalysis data and WACCM experiments. (a) reanalysis-based ensemble mean geopotential heights at 500 hPa (color shading, m) and wave activity flux (WAF) at 250 hPa (vectors, m$^2$ s$^{-2}$) of the 30 strongest negative EU months in winter (DJF) of 1951-2019 (relative to 1981-2010 climatology); (b) same as (a) but based on the SENSr2 extreme members (relative to CTRL ensemble mean); (c) same as (b) but based on the CTRL counterparts of the SENSr2 extreme members (relative to CTRL ensemble mean); (d) reanalysis-based vertical cross section of geopotential heights (color shading, m) and WAF (vectors, m$^2$ s$^{-2}$) of the ensemble mean negative EU months along the wave propagation path shown in (a); (e) same as (d) but based on the SENSr2 extreme members (relative to CTRL ensemble mean); (f) same as (e) but based on the CTRL counterparts of the SENSr2 extreme members (relative to CTRL ensemble mean). Note that the vertical components of WAF in (c)-(d) were scaled up by 200 for clear illustration. The stipples denote the 0.05 significance level.

Figure S4: 'Relative changes' to what?

Response: Here the "relative changes" are changes in terms of percentages rather than absolute values. These percentages are calculated based on the relative concentration differences in SENS extreme members using the CTRL ensemble mean concentration as benchmark. For clarification, we rephase the Fig. S4 caption to "Spatial distributions of surface $PM_{2.5}$ concentration percentage changes (unit: 100%) in extreme members of each sensitivity experiment relative to the CTRL ensemble mean result". Fig. S4c is used for direct comparison with Fig. S5 to demonstrate the effectiveness of PPI.

TECHNICAL CORRECTIONS

Page 3, line 23: Perhaps refer to 'Fig 1 (c) and (d)' instead of just 'Fig 1', since not referring to whole figure. If there are similar instances in other parts of the paper, could you perhaps change these too for clarity (e.g. page 4, line 1: 'Fig S1 (b)' rather than just 'Fig S1').

Response: Thank you for the suggestion. We change the references to specific subplots in the revised manuscript.

Page 3, line 33: I'm not sure the definition of PM_10 would be immediately obvious to all readers, although I could be wrong. Perhaps consider including a very brief definition?

Response: The definitions of $PM_{2.5}$ "(particulate matter with aerodynamic diameters of 2.5 micrometers or less)" and $PM_{10}$ "(particulate matter with aerodynamic diameters of 10 micrometers or less)" have been added after its first appearance in line 7 and line 9 of page 2.

Page 6, line 11: 'these' should be 'those'

Response: Thank you. It's changed to "those".

Page 9, line 24: 'of two indices' should be 'of the two indices'

Response: Thank you. It's changed as suggested.

Figures 3 and 7: 'inlet' should be 'inset'

Response: Thank you. All typos have been changed to "inset" in the captions of Fig. 3 and Fig. 7.

Figure 6 (a) and (b): This rainbow colour scale is not colour-blind friendly, so would be hard to interpret for some people. Perhaps use a white to blue scale, with blue indicating stronger winds?

Response: Thank you for the kind reminder. We change the color bar in Fig. 6a/b and line colors in Fig. 3 to be color-blind friendly.

Tables S3 and S4: 'MAC_Z500' in tables should be 'MCA_Z500'

Response: Thank you. The typos have been corrected.

References

IPCC, Field, C.B., Barros, V., Stocker, T.F., Qin, D., Dokken, D.J., Ebi, K.L., Mastrandrea, M.D., Mach, K.J., Plattner, G.-K., Allen, S.K., Tignor, M., and Midgley, P.M. (eds.): Managing the risks of extreme events and disasters to advance climate change adaptation. A special report of working groups I and II of the Intergovernmental Panel on Climate Change. Cambridge University Press, Cambridge, UK, and New York, NY, USA, 582 pp, 2012.

Lim, Y. K.: The East Atlantic/West Russia (EA/WR) teleconnection in the North Atlantic: climate impact and relation to Rossby wave propagation, Clim Dynam, 44, 3211-3222, 2015.

Liu, Y. Y., Wang, L., Zhou, W., and Chen, W.: Three Eurasian teleconnection patterns: spatial structures, temporal variability, and associated winter climate anomalies, Clim Dynam, 42, 2817-2839, 2014.

Sardeshmukh, P. D., and Hoskins, B. J.: The Generation of Global Rotational Flow by Steady Idealized Tropical Divergence, J Atmos Sci, 45, 1228-1251, 1988.

Screen, J.A., Deser, C., Simmonds, I. and Tomas, R.: Atmospheric impacts of Arctic sea-ice loss, 1979–2009: Separating forced change from atmospheric internal variability, Clim. Dynam., 43, 1-2, 333-344, 2014.

Simmons, A. J., Wallace, J. M., and Branstator, G. W.: Barotropic Wave-Propagation and Instability, and Atmospheric Teleconnection Patterns, J Atmos Sci, 40, 1363-1392, 1983.

Zou, Y. F., Wang, Y. H., Zhang, Y. Z., and Koo, J. H.: Arctic sea ice, Eurasia snow, and extreme winter haze in China, Sci Adv, 3, 2017.

---

## Referee Report (RR1)

**Referee report for revised Zou et al. ACP paper**

**General comments**

Thank you very much to the authors for addressing my comments. The issues I raised regarding statistical robustness have now largely been resolved, but I have a few minor comments remaining, which are listed below. Overall, I think that this paper is very interesting and would recommend it for publication, given the new mechanistic insight it provides into the important issue of haze pollution extremes in China.

**Specific comments**

Section 2.3: Thank-you for addressing my concerns about the autocorrelation. Your results presented in Tables R1 and R2, and Figures R5 and R6, convincingly show that the autocorrelation from month to month is minimal. Could you perhaps add in a sentence saying that the autocorrelation is minimal, to avoid confusion for other readers (perhaps after saying you have 90 samples on line 21 of page 6)?

Page 9, lines 35-40: With these new 95% confidence ranges from Tables S3 and S4, I am now convinced that the increase in frequency and intensity of ECP_PPI positive extremes is robust for the 30-year simulations (your Figure R3 was also useful for showing this, so thank-you for that). However, I think as well as stating that the increase in ECP_PPI extremes is statistically significant (which you have done on Page 10, lines 2-4), it should be more explicitly stated that the increase in MCA_Z500 extremes is not. This raises a bit of an issue, since the increase in ECP_PPI extremes is being attributed to the increase in MCA_Z500 extremes; if the latter isn't robust, then it's unclear whether the former is robust. However, given your Figures 4, 5, 6, and S7 in Section 3.3, there do appear to be robust effects of sea-ice loss on atmospheric circulation in the ECP region in the SENSr2 extreme members. This gives me more confidence that there is indeed a dynamical connection between SENSr2 sea-ice loss and increased ECP_PPI extremes. Perhaps this could be highlighted here?

Page 10, lines 1-2: I don't think there's enough evidence to make the following statement: 'The increase in the teleconnection pattern index MCA_Z500 is less significant than that in the regional air stagnation index ECP_PPI, suggesting a potential nonlinear relationship between large-scale circulation and regional stagnation'. In particular, as highlighted by the large 95% confidence range for MCA_Z500, there is large uncertainty in the response of MCA_Z500 extremes due to internal variability. As such, the apparent non-linear relationship between MCA_Z500 and ECP_PPI could equally be due to internal variability / chance. I therefore recommend this sentence is removed.

Section 3.3: Just a general comment - I think structuring this section around two questions has helped provide clarity, and that the addition of Figure 5c/f, Figure 6c/f/i/l, and Figure S7 has made it much more convincing that SENSr2 sea-ice loss does indeed have an effect on circulation extremes (and therefore pollution extremes) over the ECP region. This is relevant to my comment two paragraphs above.

Page 14, line 37: not sure what 'non-linear' is referring to here

Page 17, line 1: I'm confused about the phrase 'forced air stagnation response to internal variability'. How can there be a forced response to internal variability? The two are separate, although they can

resemble each other. I presume this is a typo and the response is to some forcing Callahan et al. (2019) prescribe.

**Technical corrections**

Page 2, line 34: I suggest just using 'correlation' rather than 'good correlation', as no correlation can be said to be 'good' or 'bad'. Also, I suggest splitting this sentence up as follows, as it is currently very long:

'However, a clear understanding of key dynamic processes linking complex meteorological changes to critical climate factors is still missing. This is necessary to establish a robust causal relationship between remote climate drivers and localized atmospheric responses, because a correlation does not necessarily imply causation.'

Page 14, line 20: it would help to clarify that this is for positive ECP_PPI extremes (i.e. not MCA_Z500 extremes)

Page 15, line 42: 'stratosphere' should be 'stratospheric'

Page 16, line 42: I would suggest reordering to 'Callahan et al. (2019) estimated a signal-to-noise ratio of less than one for the forced air stagnation response…'

---

## Author Response (AR2)

March 25th, 2020

Dear Dr. Maycock:

Thank you for your prompt response and positive feedback. We have further revised the manuscript according to the reviewer's comments and we would like to resubmit our paper, "Atmospheric Teleconnection Processes Linking Winter Air Stagnation and Haze Extremes in China with Regional Arctic Sea Ice Decline" (manuscript#: acp-2019-1023), for publication in the *Atmospheric Chemistry and Physics*.

Please see the point-by-point response to the review and the revised manuscript with tracked changes in this file for details. A clean version of the revised manuscript without tracked changes is also uploaded to the online submission system.

Thank you again for the time and efforts you and the reviewer put into reviewing this paper.

I look forward to hearing from you soon.

Sincerely,

Yufei Zou

Postdoctoral Research Associate
Atmospheric Sciences and Global Change Division
Pacific Northwest National Laboratory
Battelle Blvd
Richland, WA 99354
Email: yufei.zou@pnnl.gov

**Response to the referee's comments:**

General comments

Thank you very much to the authors for addressing my comments. The issues I raised regarding statistical robustness have now largely been resolved, but I have a few minor comments remaining, which are listed below. Overall, I think that this paper is very interesting and would recommend it for publication, given the new mechanistic insight it provides into the important issue of haze pollution extremes in China.

Response: Thank you very much for the comments and recommendation. We have further revised the manuscript according to your suggestions. Please see below our point-by-point responses (in blue) to your comments and corresponding changes in the revised manuscript.

Specific comments

Section 2.3: Thank-you for addressing my concerns about the autocorrelation. Your results presented in Tables R1 and R2, and Figures R5 and R6, convincingly show that the autocorrelation from month to month is minimal. Could you perhaps add in a sentence saying that the autocorrelation is minimal, to avoid confusion for other readers (perhaps after saying you have 90 samples on line 21 of page 6)?

Response: Thank you for the suggestion. We have added one sentence in line 24 of page 6 as follows: "The autocorrelation between consecutive months in each experiment is minimal, suggesting mutually independent sampling variables for statistical analysis".

Page 9, lines 35-40: With these new 95% confidence ranges from Tables S3 and S4, I am now convinced that the increase in frequency and intensity of ECP_PPI positive extremes is robust for the 30-year simulations (your Figure R3 was also useful for showing this, so thank-you for that). However, I think as well as stating that the increase in ECP_PPI extremes is statistically significant (which you have done on Page 10, lines 2-4), it should be more explicitly stated that the increase in MCA_Z500 extremes is not. This raises a bit of an issue, since the increase in

ECP_PPI extremes is being attributed to the increase in MCA_Z500 extremes; if the latter isn't robust, then it's unclear whether the former is robust. However, given your Figures 4, 5, 6, and S7 in Section 3.3, there do appear to be robust effects of sea-ice loss on atmospheric circulation in the ECP region in the SENSr2 extreme members. This gives me more confidence that there is indeed a dynamical connection between SENSr2 sea-ice loss and increased ECP_PPI extremes. Perhaps this could be highlighted here?

Response: Thank you for the comment. A previous study by Screen et al. (2014) suggested that the detection of thermodynamic responses (such as the change in Z500 in this case) requires an ensemble size approximately twice as large as thermal responses (e.g., temperature). Since PPI is partially based on near-surface temperature gradient (see Eq. (2) in the manuscript), it may be easier to detect a statistically significant response in ECP_PPI than in MCA_Z500 for the given ensemble size in this work.

We have now added a sentence with the explanation in line 4 of page 10: "In contrast, the changes of MCA_Z500 in all experiments are not statistically significant, which might be attributable to the higher difficulty and larger ensemble size requirement for detecting the dynamical-oriented responses (e.g., SLP and geopotential height anomalies) than the thermal-oriented responses (e.g., vertical temperature gradient anomalies and their effects on PPI) (Screen et al., 2014)."

Page 10, lines 1-2: I don't think there's enough evidence to make the following statement: 'The increase in the teleconnection pattern index MCA_Z500 is less significant than that in the regional air stagnation index ECP_PPI, suggesting a potential nonlinear relationship between large-scale circulation and regional stagnation'. In particular, as highlighted by the large 95% confidence range for MCA_Z500, there is large uncertainty in the response of MCA_Z500 extremes due to internal variability. As such, the apparent non-linear relationship between MCA_Z500 and ECP_PPI could equally be due to internal variability / chance. I therefore recommend this sentence is removed.

Response: Thank you for the suggestion. We have removed this sentence in the revised manuscript.

Section 3.3: Just a general comment - I think structuring this section around two questions has helped provide clarity, and that the addition of Figure 5c/f, Figure 6c/f/i/l, and Figure S7 has made it much more convincing that SENSr2 sea-ice loss does indeed have an effect on circulation extremes (and therefore pollution extremes) over the ECP region. This is relevant to my comment two paragraphs above.

Response: Thank you for the comment. These two questions were added in response to your previous concerns. We appreciate your constructive comments to improve the presentation quality and clarity of this work.

Page 14, line 37: not sure what 'non-linear' is referring to here.

Response: We removed it and rephrased the sentence as "induced atmospheric responses…" in the revised manuscript.

Page 17, line 1: I'm confused about the phrase 'forced air stagnation response to internal variability'. How can there be a forced response to internal variability? The two are separate, although they can resemble each other. I presume this is a typo and the response is to some forcing Callahan et al. (2019) prescribe.

Response: In Callahan et al. (2019), they quantify the signal-to-noise ratio of the CESM-LE historical climate by dividing the magnitude of the variable trend in the ensemble mean by the standard deviation in the magnitude of the trends across all realizations. They determine that anthropogenic forcing exerts a robust influence on the system if the signal-to-noise ratio is greater than or equal to one, following Hawkins and Sutton (2009). Therefore, the "forced air stagnation response" here denotes the change of air stagnation conditions in response to anthropogenic forcing in the CESM-LE historical realizations, which is overwhelmed by simulated natural variability given signal-to-noise ratios less than one in multiple regional air stagnation indices.

For clarification, we have rephrased the sentence in line 2 of page 17 to "Callahan et al. (2019) estimated consistent signal-to-noise ratios less than one in multiple regional air stagnation indices for Beijing based on the CESM-LE historical simulations, demonstrating the dominant role of natural variability rather than anthropogenic forcing in modulating regional circulation and ventilation".

Technical corrections

Page 2, line 34: I suggest just using 'correlation' rather than 'good correlation', as no correlation can be said to be 'good' or 'bad'. Also, I suggest splitting this sentence up as follows, as it is currently very long:

'However, a clear understanding of key dynamic processes linking complex meteorological changes to critical climate factors is still missing. This is necessary to establish a robust causal relationship between remote climate drivers and localized atmospheric responses, because a correlation does not necessarily imply causation.'

Response: Thank you for the suggestion. We have revised the sentence as suggested.

Page 14, line 20: it would help to clarify that this is for positive ECP_PPI extremes (i.e. not MCA_Z500 extremes)

Response: We clarified this in line 23 of page 14 in the revised manuscript.

Page 15, line 42: 'stratosphere' should be 'stratospheric'

Response: Thank you. It's changed to "stratospheric".

Page 16, line 42: I would suggest reordering to 'Callahan et al. (2019) estimated a signal-to-noise ratio of less than one for the forced air stagnation response…'

Response: Thank you. We have rephrased the sentence as suggested.

[revised manuscript text omitted]